# Data- and knowledge-derived functional landscape of human solute carriers

Ulrich Goldmann [1,11], Tabea Wiedmer [1,11], Andrea Garofoli [1], Vitaly Sedlyarov [1], Manuel Bichler[1], Ben Haladik [1,2], Gernot Wolf [1], Eirini Christodoulaki [1], Alvaro Ingles-Prieto [1], Evandro Ferrada [1], Fabian Frommelt [1], Shao Thing Teoh [1], Philipp Leippe [1], Gabriel Onea [1,3], Martin Pfeifer [4], Mariah Kohlbrenner [4], Lena Chang[4], Paul Selzer [4], Jürgen Reinhardt [4], Daniela Digles [5], Gerhard F Ecker [5], Tanja Osthushenrich [6], Aidan MacNamara [6], Anders Malarstig [7], David Hepworth [8] & Giulio Superti-Furga [1,9,10]✉

## Abstract

**The human solute carrier (SLC) superfamily of ~460 membrane transporters remains the largest understudied protein family despite its therapeutic potential. To advance SLC research, we developed a comprehensive knowledgebase that integrates systematic multi-omics data sets with selected curated information from public sources. We annotated SLC substrates through literature curation, compiled SLC disease associations using data mining techniques, and determined the subcellular localization of SLCs by combining annotations from public databases with an immunofluorescence imaging approach. This SLC-centric knowledge is made accessible to the scientific community via a web portal featuring interactive dashboards and visualization tools. Utilizing this systematically collected and curated resource, we computationally derived an integrated functional landscape for the entire human SLC superfamily. We identified clusters with distinct properties and established functional distances between transporters. Based on all available data sets and their integration, we assigned biochemical/biological functions to each SLC, making this study one of the largest systematic annotations of human gene function and a potential blueprint for future research endeavors.**

**Keywords** Human Gene Function; Knowledgebase; Membrane Transporters; Multimodal Data Integration; Solute Carriers
**Subject Categories** Membranes & Trafficking; Methods & Resources

See also: G Wolf et al, T Wiedmer et al, F Frommelt et al

## Introduction

Systematically assessing human gene function is an important but daunting challenge in any defined biological system. The RESOLUTE and REsolution consortia mounted a concerted action to tackle the solute carrier (SLC) superfamily of membrane transporters, a subsystem thought to be approachable by size (~460 genes), of coherent biochemical role and related as well as interconnected functions, spanning the entire functional space of secondary active transport of small organic and inorganic molecules. SLCs have a special role in pharmacology as they include targets of some very prominent drugs, like gliflozins and monoamine uptake inhibitors (Wang et al, 2020; Galetin et al, 2024), contain many disease-linked genes (Lin et al, 2015; Wang et al, 2020; Schlessinger et al, 2023), and fulfill important functions in drug uptake and disposition (Girardi et al, 2020; Giacomini et al, 2010). It is not surprising, therefore, that interest around SLC transporters has increased in the pharmaceutical industry (César-Razquin et al, 2015). The RESOLUTE consortium of the Innovative Medicine Initiative of the European Union was a public–private partnership that started in 2018 involving seven pharmaceutical companies and six academic institutions aimed at broadly unlocking solute carriers as a target class. The goal was to provide research tools and advance the knowledge for as many human SLCs as possible, thus lowering the entry barrier for further research projects and drug discovery (Superti-Furga et al, 2020); see also https://re-solute.eu/communications for a detailed account of the project. In 2021, the REsolution consortium formed with nine partners to extend the scope of RESOLUTE by anchoring its results in a biomedical context (Wiedmer et al, 2022). The challenges were manifold. How to tackle the experimental functional annotation of a group of genes of such size and scope? How to optimize knowledge advancement in the most efficient way? How to integrate data to achieve knowledge beyond the sum of the parts?

[1]CeMM Research Center for Molecular Medicine of the Austrian Academy of Sciences, Vienna, Austria. [2]St. Anna Children's Cancer Research Institute, Vienna, Austria. [3]Department of Pediatrics and Adolescent Medicine, Medical University of Vienna, Vienna, Austria. [4]Novartis Pharma AG, Basel, Switzerland. [5]University of Vienna, Department of Pharmaceutical Sciences, Vienna, Austria. [6]Bayer AG, Leverkusen, Germany. [7]Pfizer Research and Development, Stockholm, Sweden. [8]Pfizer Research and Development, Cambridge, MA, USA. [9]Center for Physiology and Pharmacology, Medical University of Vienna, Vienna, Austria. [10]Fondazione Ri.MED, Palermo, Italy. [11]These authors contributed equally: Ulrich Goldmann, Tabea Wiedmer. ✉E-mail: gsuperti@cemm.oeaw.ac.at

Solute carriers are a superfamily of transporters, arbitrarily grouped to rationalize the genetic nomenclature for multipass transmembrane carriers that are not ATPases, aquaporins or channels (Hediger et al, 2004). SLCs localize to the plasma membrane and different intracellular membranes (Pizzagalli et al, 2021). They have been classified into 70 families, with family members sharing more than 20% sequence identity (Hediger et al, 2004). Recently, the development of deep-learning methods such as AlphaFold allowed to discern about 20 different structural folds to which the families can be allocated (Ferrada and Superti-Furga, 2022; Xie et al, 2022). Many of these folds are evolutionarily ancient, reflecting their fundamental cellular and physiological roles (Höglund et al, 2011). As a result of different rounds of gene duplication, paralog genes have contributed to extensive functional redundancy. This is part of the reason why assigning specific functions to single SLC genes has not been straightforward, and a good proportion of SLC genes could be termed 'orphan' (César-Razquin et al, 2015; Meixner et al, 2020). SLCs are expressed across all human tissues, but their specificity of expression varies widely: while some SLCs are expressed only in very specific tissues, others are ubiquitously expressed across all tissues (César-Razquin et al, 2015; Zhang et al, 2019; O'Hagan et al, 2018). Each cell and tissue expresses around half of all SLCs, which is thought to reflect the metabolic requirements of that particular tissue (César-Razquin et al, 2015; O'Hagan et al, 2018). Of these, only between 20-30 SLCs are deemed essential (Wang et al, 2015; Blomen et al, 2015). The 13 glucose transporters within the SLC2 and SLC5 families provide an exemplary case of how the systemic redundancy, discrepancy in essentiality, and sheer number of SLC genes challenge traditional cell-biological and biochemical approaches to gene function assignments.

It can be expected that most small molecules transported by the ~460 members of the SLC superfamily are endogenous metabolites, ranging from inorganic ions to complex prosthetic groups. However, some transporters may have evolved to transport environmental components that were endogenously utilized in the past but have turned into xenobiotics (Gründemann et al, 2005). Nevertheless, the chemical space for which transport has evolved is finite, with the Human Metabolome Database (https://hmdb.ca) currently listing ~220,000 chemical species. The number of genes encoding proteins responsible for the active transport of molecules is relatively small, estimated to be under a thousand (Elbourne et al, 2017). SLCs represent the largest group among these. From these considerations, the problem of assigning functions to each of the SLC genes is one of matchmaking. Such an SLC-solute matchmaking process can be facilitated by exclusion, systematically eliminating unlikely pairings based on experimental data, known gene characteristics (also from other organisms), and functional or structural constraints. By ruling out non-functional or less likely associations, the set of possible gene-function matches is progressively narrowed down, ultimately leading to an optimal mapping where each gene is matched to its most likely or confirmed function. Such an approach relies on both the inclusion of positive matches and the exclusion of incompatible or improbable ones, ensuring a precise and accurate assignment of functions to genes.

Thus, the effort of creating a functional landscape is essentially optimizing a virtual functional distance between SLC genes using data modalities we obtained in the RESOLUTE and REsolution efforts. Among the experimental data modalities, targeted metabolomics provides the most direct assessment of substrate transport and its implications on metabolic consequences (Wiedmer et al, 2025). This data is then complemented with transcriptomics to reflect the cellular functional state (Wiedmer et al, 2025) and with interaction proteomics to provide the protein environment as a molecular basis for functional states (Frommelt et al, 2025). Knowledge of the subcellular localization of SLCs further complements the understanding of their cellular environment. In this study, we obtained experimental evidence of subcellular localization for the entire SLC superfamily. Beyond experimental data, we also integrated publicly available knowledge of both physical and physiological parameters such as structural prediction, tissue expression, substrate annotation, and disease association to further characterize SLC function.

Integration of biomedical, multimodal data sets remains challenging (Cai et al, 2022). A number of methods have been developed, each with specific requirements and applications. Multi-omics factor analysis (MOFA) (Argelaguet et al, 2018) and MultiMAP (Jain et al, 2021) are tailored for multi-omics data. They do not require complete data, but they will not work with qualitative data such as disease associations or lists of annotated substrates. Translating these annotations to quantitative similarities allows for the application of different integration algorithms such as similarity network fusion (SNF) (Wang et al, 2014) or AlignedUMAP (Dadu et al, 2023). They do, however, require complete data and will not work for SLCs missing any annotation or experimental data. In the present study, we propose and apply a new approach to integrate the diverse data sets collected into a functional landscape of human SLCs.

It should be noted that there is no single right solution, as there are many more transported molecules than transporters, and the degree of redundancy in the SLC-solute matchmaking increases with our knowledge of them. Accordingly, any concept of "deorphanisation", familiarly used in reference to nuclear hormone receptors and GPCRs, would be misleading in the case of SLC transporters. How then can a function of an SLC gene be assigned in an unequivocal way? The present study represents such an effort.

## Results

### Characterization of the human SLC superfamily members by their structure, expression, and substrates

To start a systematic assignment of biochemical and biological properties to each member of the solute carrier superfamily demanded a definition of the criteria used to include or exclude genes from the analysis. To date, the HUGO Gene Nomenclature Committee (HGNC) has included 449 human genes in the solute carrier superfamily (Seal et al, 2023). An additional fourteen genes were suggested by Perland and Fredriksson and another one by Ferrada and Superti-Furga (TMEM144) (Perland and Fredriksson, 2017; Ferrada and Superti-Furga, 2022), making up a total of 464 human genes (Dataset EV1). These included eight pseudogenes, two insertases (MTCH1 and MTCH2), and three auxiliary subunits (SLC3A1, SLC3A2, SLC51B) without transport function. Based on the sequence similarity of SLCs, the superfamily was divided into 70 families (Perland and Fredriksson, 2017; Hediger et al, 2004). At

the start of the RESOLUTE consortium, a slightly lower number of proteins were classified as SLCs, and therefore only 447 SLCs of 69 families were considered for the collection of data in this and accompanying studies (Fig. 1A) (Wiedmer et al, 2025; Frommelt et al, 2025; Wolf et al, 2025).

Based on experimental structures and structures modeled using AlphaFold, we have recently identified 24 distinct transmembrane structural folds within the SLC superfamily (Ferrada and Superti-Furga, 2022). Inspired by the description and illustration of the human kinome (Manning et al, 2002), a similarly large and important protein superfamily, we constructed an unrooted tree, clustering the SLCs based on their structural similarity. This tree consisted of five major branches. Four of these were the MFS, LeuT, MitC, and DMT clades, each with more than 30 members of a common structural fold and collectively covering two-thirds of all SLCs ($n = 301$; 67.3%). The fifth branch, covering one-third of the SLCs ($n = 146$; 32.7%), was much more heterogeneous and represented 20 more structural folds with fewer than 25 members each and also quite different numbers of transmembrane helices (UraA, Glt, NhaA, ZIP, SLC64, IT, MnN3, SLC51, SLC53, SLC44, SLC56, AmtB, SLC41, PiT, MATE, NPC1, CNT2, CNT1, YiiP, NCX) (Fig. 1B). Like the kinome tree, the SLC tree offered itself as a visualization platform for a variety of SLC properties and data useful to be annotated from an evolutionary structural perspective, such as substrate preference. A web-based version allows the upload and overlay with any gene-level annotation and interactive navigation down to individual SLCs (https://re-solute.eu/resources/dashboards/slctree).

In addition to the variety of structural folds, the SLC superfamily is also characterized by various patterns of expression. The Human Protein Atlas provided cell line and tissue expression data for 455 human SLCs (Fig. 1C) (Uhlén et al, 2015). Among the 1206 cell lines analyzed, an average of 2.2% of the expressed genes were SLCs. An individual cell line expressed 256 SLCs (56.2%) on average (Fig. EV1A). Similarly, in the 50 tissues analyzed, an average of 2.2% of the expressed genes were SLCs, with tissues expressing an average of 300 SLCs (65.9%) (Fig. EV1B). 'Tissue-elevated' gene expression was defined by the Human Protein Atlas as a gene having elevated expression levels in a single tissue or a group of tissues (Fig. EV1C). The group of brain tissues showed the highest number of tissue-elevated SLCs ($n = 72$), followed by liver ($n = 62$), kidney ($n = 53$), the group of intestine tissues ($n = 47$), testis ($n = 31$), retina ($n = 25$), and skeletal muscle ($n = 21$) (Fig. 1D).

The complex structural categorization and the diverse expression patterns within this superfamily are likely rooted in the ability of SLCs to transport a wide array of substances across different cellular membranes. We previously generated a comprehensive overview of substrates of the entire superfamily through manual annotation based on literature (Meixner et al, 2020). Considering recent advances in the field and limitations of the previous annotation, we updated and refined the annotation, as well as the inclusion criteria and the mapping process (Methods). The updated annotation consisted of 2044 SLC-compound-publication pairs, assigning 468 distinct substrates to 329 SLCs based on 678 publications, with a median of 3 substrates transported per SLC (Dataset EV2). Nearly half of the 468 distinct compounds ($n = 213$; 45.5%) were transported by only a single SLC, while another third ($n = 164$, 35.0%) were annotated to three or more transporters.

Inversely, a single substrate was annotated for about a quarter of these SLCs ($n = 88$; 26.7%), while 36 SLCs (10.9%) were annotated with more than ten substrates each. Among these, SLC22A1 had the most annotations with 39 substrates. This highlights the heterogeneity in both substrate specificity and depth of knowledge across the SLC superfamily. A quarter of all SLCs ($n = 110$) remained without an annotated substrate (i.e., orphan) (Fig. 1E). Compared to our previously published annotation (Meixner et al, 2020), 23 SLCs now have substrates assigned and another 13 SLCs with previously annotated substrates were now classified as orphans since existing evidence did not fulfill our set criteria (Fig. EV1D). A recent study (Zhang et al, 2025) annotates substrates for an additional 19 SLCs, although most of these annotations are inferred from orthologs and do not meet our criteria (Methods). For completeness, these additional annotations for a total of 32 SLCs are included in Dataset EV2.

For further analyses of SLC substrate specificity and to facilitate integration with other data sets, we established a single-label substrate classification per SLC. We defined a set of nine biologically and biochemically relevant substrate classes based on a manual selection of higher-level ChEBI terms (Hastings et al, 2016): "amino acids, peptides and derivatives"; "carbohydrates and derivatives"; "carboxylic acids and derivatives"; "heteroarenes"; "inorganic ions"; "lipids and steroids"; "nucleosides, nucleotides and nucleotide-sugars"; "organic ions"; "transition element cations". Annotated substrates were matched to these classes via the ChEBI ontology tree. Substrates that matched different classes were semi-automatically assigned to a single class. Similarly, SLCs with multiple substrates of different classes were assigned to a single class (Methods). This ontology-based classification approach allowed us to divide the largest substrate class "other" from the previous classification (Meixner et al, 2020) into biologically and biochemically relevant substrate classes (Fig. EV1D). In addition, SLCs from the previous classes "ion" and "metal" were split into more precisely termed classes such as "carboxylic acids and derivatives", "organic ions", "inorganic ions", and "transition element cations". Orphan SLCs remained the largest class with 110 SLCs. Among the 329 substrate-annotated SLCs, "amino acids, peptides and derivatives" was the largest substrate class (75 SLCs; 22.8%), followed by "inorganic ions" (50 SLCs; 15.2%) and "organic ions" (43 SLCs; 13.1%) (Fig. 1F). Visualization on the SLC tree demonstrated only partial overlap of SLC substrate classes to structural folds, where some folds were specific to a substrate class (e.g., transition element cations transporters in YiiP fold) and others were not (e.g., SLCs of five different substrate classes in UraA fold) (Fig. 1G).

## Survey of SLC superfamily-wide disease associations

The diversity in substrates of the SLC superfamily and the resulting linkage to many metabolic processes imply a broad role in physiology in general and human health in particular. Many SLCs are linked to Mendelian diseases, even more are associated with a large variety of human disease traits, and several members were shown to be successful drug targets (Lin et al, 2015; Wang et al, 2020; Schlessinger et al, 2023). To obtain a comprehensive overview of the current knowledge of SLC genetics and its impact on human biology, we performed programmatic data mining to collect and curate data from a wide range of open-source databases (Fig. 2A).

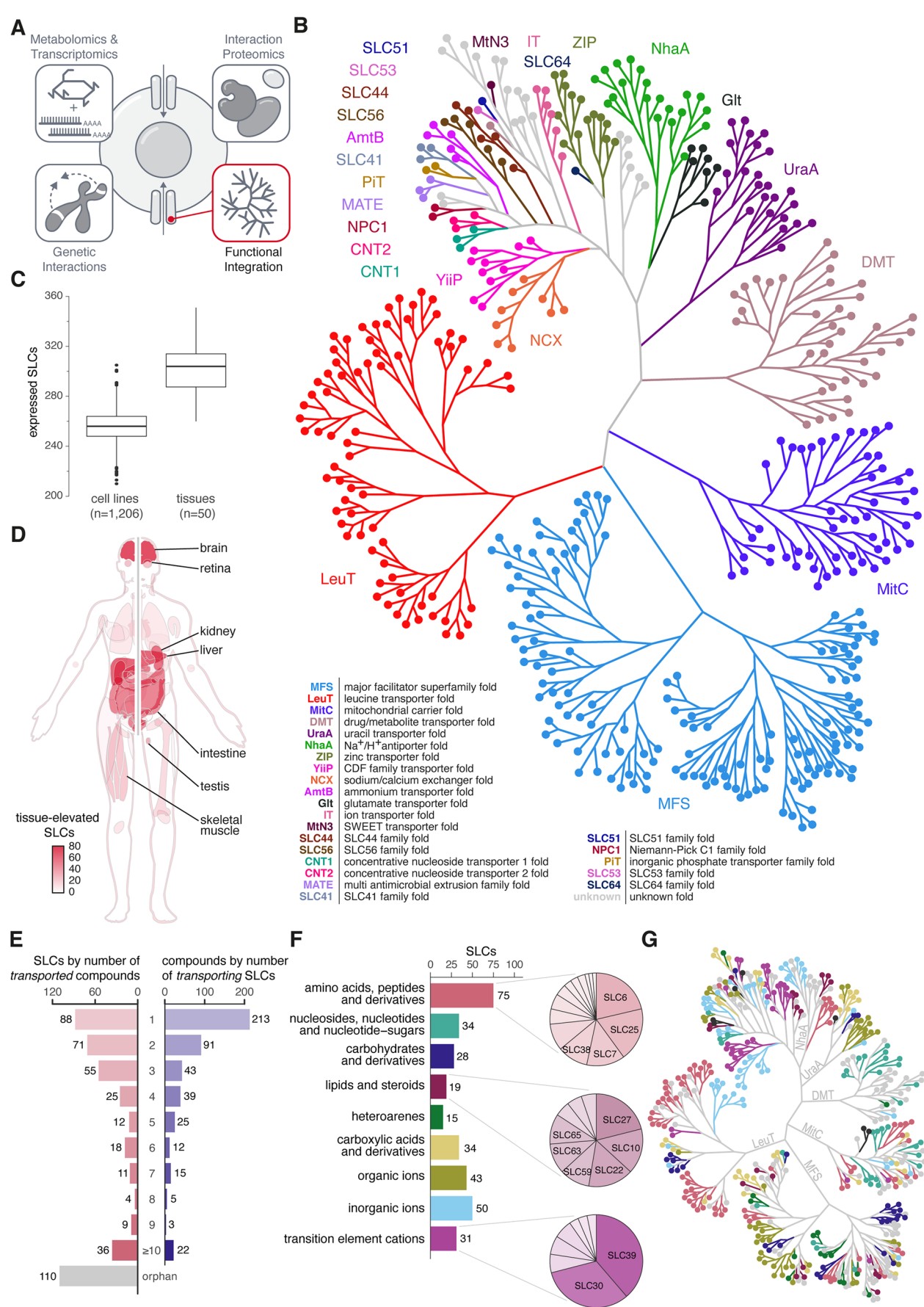

**Figure 1.   Overview of the solute carrier superfamily.**

(A) Overview representation of the SLC superfamily-wide RESOLUTE paper collection. (B) Unrooted tree representation of the 447 SLCs included in this study, based on experimental and modeled structures (data from (Ferrada and Superti-Furga, 2022)). The SLC tree illustrates 24 folds based on their similarity in different colors. A web-based version allows its exploration down to individual SLCs as well as uploading and overlaying with gene-level annotations (https://re-solute.eu/resources/dashboards/slctree). (C) Distribution of expressed SLC genes within 1206 cell lines and 50 tissues (data from (Uhlén et al, 2015)) represented as boxplots. The boxes and the lines within show the interquartile ranges and the medians, respectively. Whiskers extend to values within 1.5 times the interquartile ranges and values beyond are displayed as points. Figure EV1A,B shows the number of expressed SLC genes also in relation to the total number of expressed genes in a particular cell line or tissue, respectively. (D) Schematic representation of a human body where tissues are colored by the number of SLCs with a corresponding tissue-elevated expression pattern. Tissues featuring more than 20 SLCs are indicated. Figure EV1C shows the data for all tissues. (E) Summary of literature-based substrate annotation, demonstrating the redundancy of compounds transported per SLC as well as the redundancy of SLCs transporting a specific compound. The left chart shows the number of SLCs annotated to transport from 1 up to 10 or more substrates, and SLCs without annotated substrates (i.e., orphan). The right chart shows the number of compounds annotated to be transported by 1 up to 10 or more SLCs. (F) Bar chart with the number of SLCs classified into nine substrate classes. Pie charts illustrate the proportion of families within SLCs annotated as "amino acids, peptides and derivatives", "lipids and steroids", and "transition element cations" transporters. (G) Substrate class for each SLC illustrated on the SLC tree. Color code according to substrate class as in (F).

Selected databases included genetic and clinical evidence (UniProt, Ensembl, gnomAD, ClinVar, Orphanet) and statistically processed data (IEU OpenGWAS project, Genebass, Open Targets, Priority index) (Table EV1).

Our methodical approach resulted in curated data for 456 members of the SLC superfamily, offering a unified repository of information previously dispersed across various resources. We found 7,947,666 variants mapped to human SLC genes as well as various genetic and phenotypic annotations. Many of these variants are neutral, but they can also be deleterious through various types of mechanisms, including direct impact on gene expression, protein structure or function. A protein-altering consequence on the canonical transcript of an SLC gene was annotated for 98,655 of those variants. Missense variants, which are single-nucleotide mutations translating into single amino acid substitutions, made up 90.1% of all the collected protein-altering variants (Fig. EV2A).

The traits found associated with SLCs included human diseases, symptoms, biological quantitative measurements, and general phenotypes not directly linked to pathogenicity. Associated traits were curated to a standardized terminology to eliminate redundancies, clarify ambiguous terms, and distinguish disease-related from non-disease traits. We implemented a workflow to map terms to community-established ontologies, employing Monarch Disease Ontology (Mondo) (Vasilevsky et al, 2022) for disease associated terms and the Experimental Factor Ontology, Human Phenotype Ontology, or Gene Ontology for broader biological terms (Malone et al, 2010; Köhler et al, 2021; Ashburner et al, 2000). The process of standardization and ontology mapping enabled us to annotate previously unclassified variants as likely pathogenic due to their association with disease traits (Fig. 2A).

We found a total of 7816 trait associations among the protein-altering variants across 207 SLCs. Of these associations, 3244 (41.5%) corresponded to 2402 variants and were classified as pathogenic. Moreover, 485 trait associations for 202 SLCs emerged from bundled missense and loss-of-function gene-level tests for rare variants, of which 216 (44.5%) were classified as pathogenic. Overall, we collected and mapped 780 distinct associations of pathogenic traits to 228 SLCs (Fig. EV2B; Dataset EV3), while another 228 SLCs were still without disease association, resulting in an average of 1.7 pathogenic traits across all 456 SLCs studied (Fig. 2B). For 125 SLCs, all disease associations were already clinically established, i.e., reported in the ClinVar or Orphanet data sets. We found new likely pathogenic associations for 103 SLCs, of which 35 SLCs already had at least one other clinically established

association and 68 SLCs had no previous association in ClinVar or Orphanet at all (Fig. 2C). Among the 127 likely pathogenic terms associated with those 68 SLCs (Fig. EV2C), we found systemic lupus erythematosus (SLE) associated with SLC15A2, SLC15A4, and SLC17A3. Although the involvement of SLC15A4 in the development of SLE has been well-described (Kobayashi et al, 2014; Heinz et al, 2020), these missense variants were not yet classified as pathogenic in ClinVar and Orphanet. No reports exist to date for the pathogenicity of SLC15A2 and SLC17A3 in SLE. Furthermore, type II diabetes was associated with SLC16A11, SLC30A8, SLC39A11 and MTCH2. The associations of SLC30A8 and SLC16A11 with diabetes are well-described, but the disease mechanisms have not been fully elucidated yet (Krentz and Gloyn, 2020; Rusu et al, 2017; Hoch et al, 2019) and accordingly ClinVar and Orphanet do not report this association. MTCH2 was among the genes with many novel likely pathogenic associations, which have been previously reported in genome-wide association studies focused on obesity and diabetes (Kang et al, 2020). In contrast, we found no literature about the role of SLC39A11 in type II diabetes.

The number of associated disease terms was not correlated with the number of pathogenic variants (Fig. 2D). Most of the SLCs associated with a high number of diseases had a low number of pathogenic variants: SLC39A8, SLC44A4, SLC17A1, SLC22A1, SLC22A4, and SLC44A2 were each associated with a high number (>20) of diseases, but those associations are originating from a low number (<10) of variants. SLC39A8 is a well-documented case of a pleiotropic gene (i.e., one gene associated with multiple phenotypes), for which only six pathogenic protein-altering variants have been linked to 44 different disease terms in humans (Pickrell et al, 2016). Conversely, there are genes like SLC26A4 and SLC22A5 with more than 100 variants each but resulting in associations with only a few diseases (6 and 3, respectively). The SLCs with a high number of pathogenic variants did not appear to be from any specific families or structural folds (Fig. 2E).

To provide an overview of the implications of the superfamily's disease associations, we mapped them to different disease areas employing the Mondo ontology (Methods). An SLC was associated with about two disease areas on average. The disease areas with the most SLCs associated were "Mendelian disease" (164 SLCs), "metabolic disease" (116 SLCs) and "nervous system disorder" (90 SLCs) (Fig. 2F). Only a few disease areas were adequately covered by potential tool compounds for their associated SLCs based on the recently published list for 51 SLCs (Digles et al, 2024).

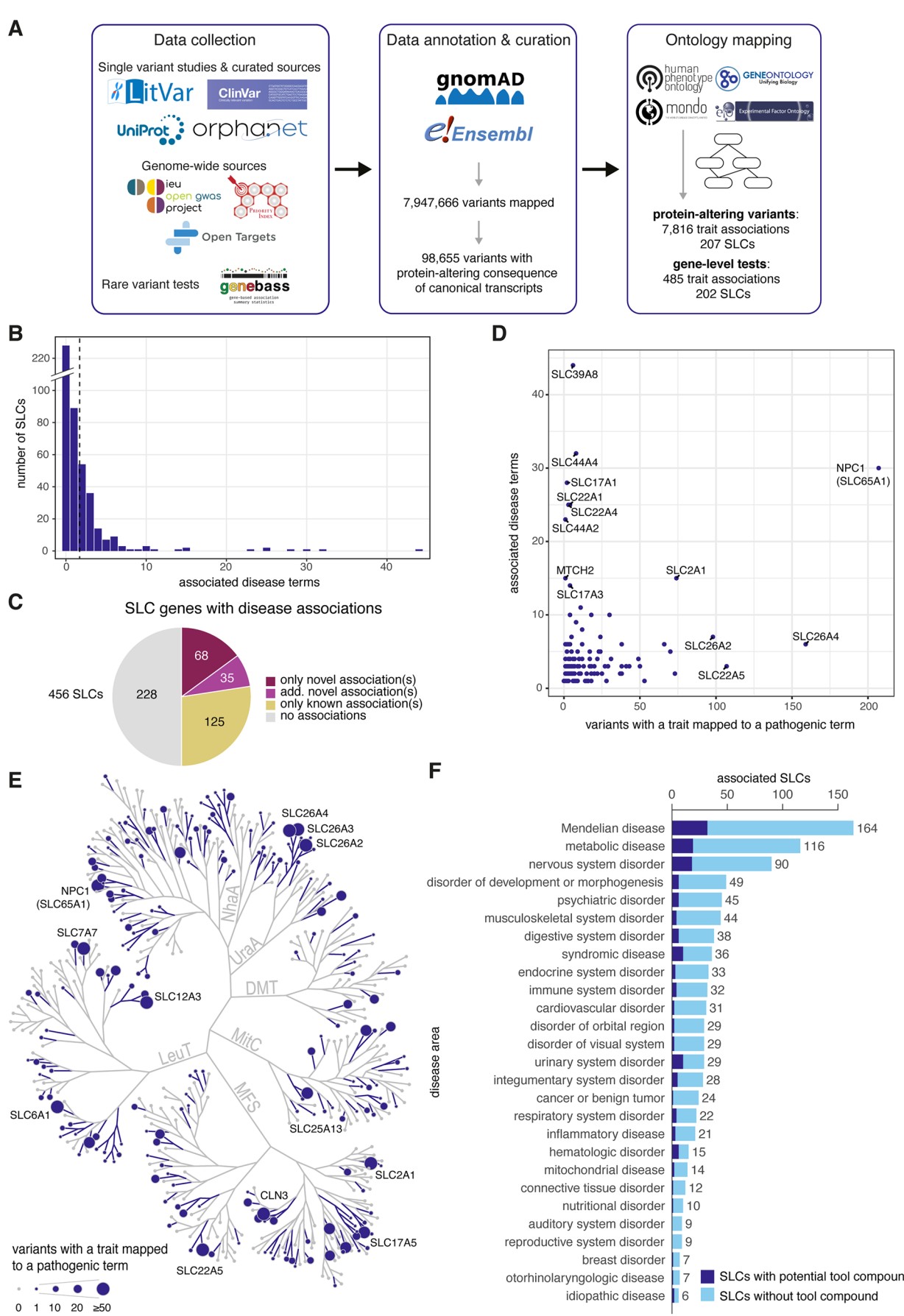

**Figure 2.    Collection of SLC genetic variants and disease associations.**

(A) Overview of workflow, consisting of data collection, data annotation & curation, and ontology mapping. (B) Histogram showing the distribution of SLC by their associated disease terms. The average of 1.7 disease associations is indicated by the dashed line. (C) Proportion of SLC genes with disease associations when comparing the current data set to previous pathogenic associations in ClinVar and Orphanet. "Only novel associations": no previous pathogenic associations in respective sources; "additional novel associations": novel associations in addition to previously reported pathogenic associations in respective sources; "only known associations": our data set includes the same pathogenic associations as previously reported by respective sources. (D) Number of distinct associated diseases compared to the overall number of pathogenic variants in each SLC. (E) The number of pathogenic variants per SLC highlighted on the SLC tree. SLCs with more than 40 variants with a trait mapped to a pathogenic term are labeled. (F) Distribution of pathogenic associations of SLCs across main disease areas (≥5 associated SLCs). Disease areas are defined by the child terms of the Mondo term "Human disease" (Methods). Only the disease areas "hematologic disorder" and "urinary system disorder" have more than one-third of their associated SLCs covered by potential tool compounds, as defined in (Digles et al, 2024).

## Experimental annotation of subcellular location for the SLC superfamily

SLCs are localized at different organellar membranes and contribute to the compartmentalization of metabolism. Knowledge of their subcellular localization is crucial for understanding their function and is, to date, rather broadly annotated in public databases. However, those data sets have limitations and caveats, such as annotations based on sequence similarity or antibody-based experimental evidence, even though antibodies often lack a thorough validation of their target specificity (Ayoubi et al, 2023). Therefore, we generated, collected and curated experimental evidence for each member of the SLC superfamily.

We processed 560 Jump-In T-REx HEK293 cell lines, each expressing an SLC fused to an HA-tag under a doxycycline-inducible promoter, followed by an immunofluorescence-based high-content imaging approach (Fig. 3A). Of those, 447 cell lines expressed C-terminally HA-tagged SLCs and 113 N-terminally HA-tagged SLCs, the latter due to unsatisfactory C-terminal tagging. SLC expression was induced by doxycycline, followed by a washout period and subsequent immunofluorescent staining with HA and compartment markers for the cytoplasm, plasma membrane, mitochondria, lysosome, endosome, endoplasmic reticulum (ER), Golgi, peroxisome, cytoskeleton and the nucleus. Using high-content imaging, we acquired at least 768 images per cell line which were SLC- and compartment-wise individually levelled and then visually inspected. We generated a manual assessment of ten consolidated locations (ER; Golgi; plasma membrane; perinucleic region; lysosomes; endosomes; mitochondria; cytosol; cytoskeleton; lamellipodia) for 407 SLCs that were adequately expressed. Each assessment contained, for each compartment, (i) a signal score representing the approximated relative fluorescence intensity compared to the total fluorescence intensity, and (ii) a confidence value ranging from 1 (very weak) to 5 (very high). The annotations were carefully curated in several rounds to minimize human bias and provide a ground truth for the development of machine learning-based localization prediction. The last curation rounds focused on studying mismatches between prediction and human assessment, leading to an improved human statement as well as improved predictive models (Baranowski et al, in preparation).

To compare and complement our experimental results and annotations of subcellular localization with publicly available sources, we retrieved respective data sets from the Human Protein Atlas (HPA; version 23.0) (Thul et al, 2017), Gene Ontology (GO) (Gene Ontology Consortium et al, 2023) and UniProt (UniProt Consortium, 2023). The data sets varied largely in the number of terms, the hierarchy between the terms and their reliability scores:

HPA provided 40 terms structured in a simple hierarchy with four levels of reliability, the GO "cellular component" subset consisted of 4058 terms in an ontology graph with 26 evidence codes, and UniProt used 542 subcellular location terms in a simpler ontology graph and four evidence codes. We first mapped and aligned the three public data sets and our data set to a total of ten subcellular location terms: endoplasmic reticulum, Golgi apparatus, plasma membrane, nucleus, lysosome, endosome, mitochondrion, cytosol, cytoskeleton, and cell projection (Methods; Fig. EV3A). The RESOLUTE data set included the highest number of 1107 annotations for 403 SLCs, followed by 741 annotations for 409 SLCs by GO, 469 annotations for 355 SLCs by UniProt, and 339 annotations for 231 SLCs by HPA. An overlap of 178 SLCs was covered by all four sources, and another 146 SLCs by RESOLUTE, GO, and UniProt (Fig. 3B).

A considerable number of these annotations were of weaker confidence, labeled as uncertain, based on computational prediction or author/curator statements without experimental evidence. As the combination of annotations from different sources increases the false positive rate, we stringently filtered the annotations to experimentally derived, high-confidence localization annotations (Methods). This high-confidence subset encompassed 434 annotations for 340 SLCs by RESOLUTE, 469 annotations for 300 SLCs by GO, 312 annotations for 247 SLCs by UniProt, and 284 annotations for 213 SLCs by HPA (Fig. 3C). Merging the four high-confidence subsets resulted in 873 localization annotations for 418 SLCs and an average of about 2 locations per SLC (Dataset EV4). 487 location annotations (55.8%) were supported by a single source, 386 annotations by multiple sources (44.2%), with only 52 annotations (6.0%) supported by all four data sources (Fig. 3D). All sources reported the plasma membrane as the most frequent location of SLCs, ranging from close to 100 SLCs in HPA to almost 200 SLCs in GO. Up to seven different locations were annotated for individual SLCs, with a similar distribution of the number of locations per SLC across all sources (Fig. EV3B).

To judge the level of agreement on the high-confidence annotations between the four data sources, we treated the localization annotation as a multi-label classification problem. We calculated, for each pair of data sets, the precision and recall per location and combined those metrics into an overall accuracy score across all locations (micro-averaged F1-score (Takahashi et al, 2022)). We found the overall highest score, and therefore agreement, between GO and UniProt annotations. The RESOLUTE data set agreed equally well with both UniProt and GO annotations and to a lesser extent with HPA, which generally showed the least agreement with all other three annotations (Fig. 3E). Comparing the annotation sources for each localization, we found the highest

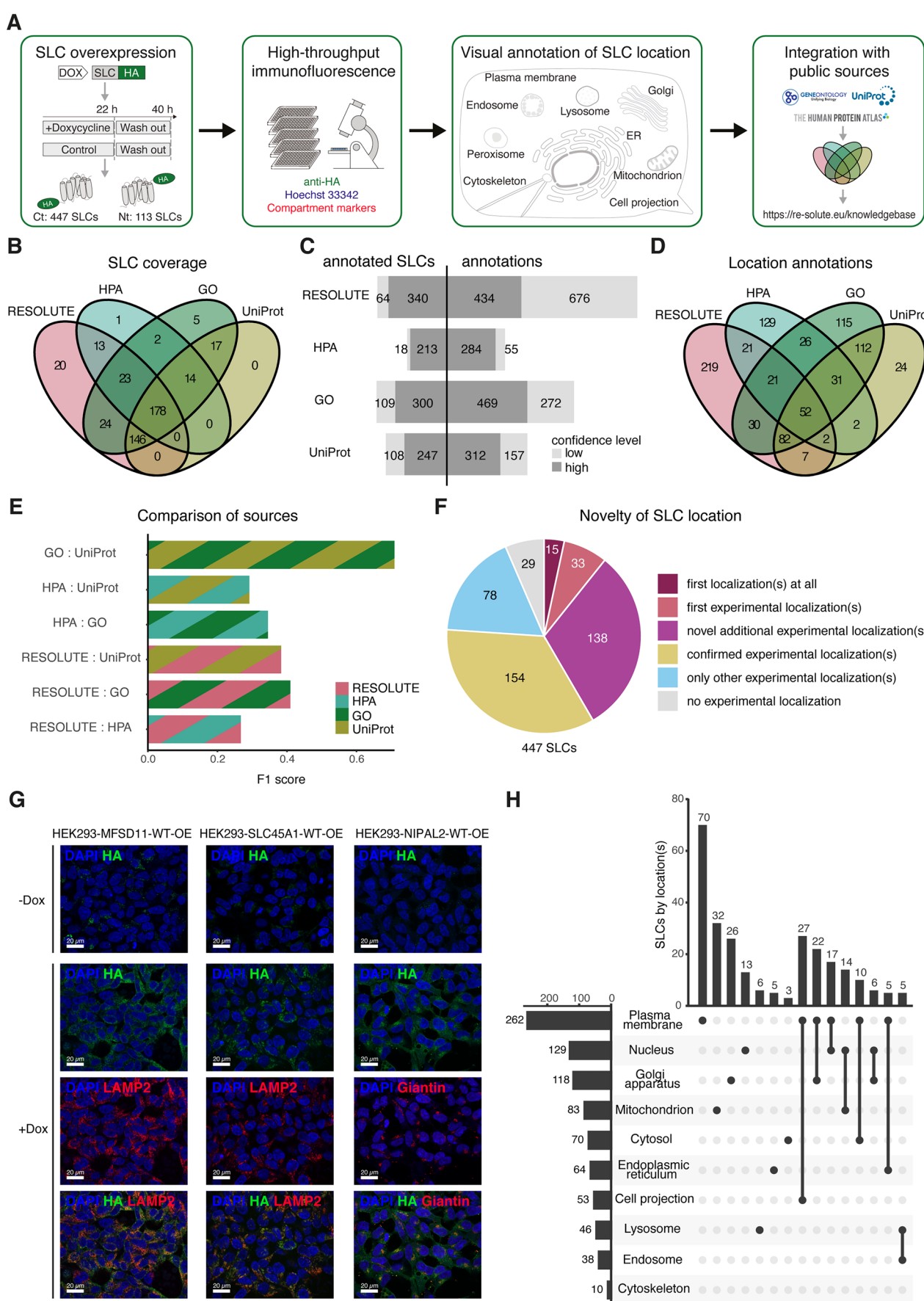

**Figure 3.    Subcellular localization of the SLC superfamily.**

(A) Workflow for the generation of a subcellular localization data set by experimental high-throughput immunofluorescence combined with integration of public data. (B) Comparison of SLC coverage between the acquired data set (RESOLUTE) and public sources Human Protein Atlas (HPA), Gene Ontology (GO), and UniProt. (C) Numbers of total and high-confidence annotated SLCs and localization annotations by RESOLUTE, HPA, GO, and UniProt. For filtering criteria, please refer to Methods. (D) Comparison of the overlap between high-confidence localization annotations in RESOLUTE, HPA, GO, and UniProt. (E) Scoring of the overall agreement in high-confidence subcellular localization annotations for each pair of annotation sources. The F1-score combines precision and recall for each localization, with a higher score corresponding to higher consistency. (F) Novelty of high-confidence localization annotation per SLC by the RESOLUTE data set compared to HPA, GO, and UniProt. (G) Immunofluorescence images of cell lines HEK293-MFSD11-WT-OE, HEK293-SLC45A1-WT-OE, and HEK293-NIPAL2-WT-OE. Cells were treated with doxycycline for 22 h followed by an 18 h washout. Green channel: HA-tag of overexpressed SLC. Red channel: compartment marker for lysosomes (LAMP2) or Golgi (Giantin). Blue channel: Hoechst 33342 for nucleus. Images were acquired using a ×60 objective. Scale bars indicate 20 μm. (H) UpSet plot (Conway et al, 2017) visualizing the number of SLCs annotated for each location (horizontal bars) and number of SLCs annotated for specific location combinations (vertical bars) in the consolidated high-confidence data set with combined evidence from RESOLUTE, HPA, GO, and UniProt. For readability, the plot displays only SLCs that are either exclusively localized or found in combinations of two locations with at least five SLCs. The full localization data set is available in Dataset EV4.

agreements for mitochondrion, plasma membrane and Golgi. In contrast, lower agreement was observed for ER location (Fig. EV3C).

Assessing the novelty of high-confidence experimental subcellular location annotations, the RESOLUTE data set confirmed the previously experimentally derived annotations for approximately one-third of the SLC family (154 SLCs; 34.5%). For another third (138 SLCs; 30.9%), RESOLUTE provided experimental evidence for locations in addition to previously reported ones. Among those, Golgi, nucleus and cell projection locations were most frequent, which might be influenced by the tagged overexpression of the SLC or by the inherently subjective manual annotation (Fig. EV3D). RESOLUTE reported the first experimental annotations for 48 SLCs (10.8%), including 15 SLCs that contained no prior location annotations at all (Fig. 3F). Among those, we found the glucose transporter SLC45A1 and the orphan MFSD11 localized at the lysosome, as well as the orphan transporter NIPAL2 at the Golgi and plasma membrane (Fig. 3G).

In the consolidated data set from public sources and our experimental work, more than half of the SLCs ($n = 262$; 58.6%) were annotated at the plasma membrane, but only 70 of them exclusively (Fig. 3H). 155 SLCs (34.7%) had one location, and 139 SLCs (31.1%) had two locations annotated. The most common pair of locations was the plasma membrane and cell projection (27 SLCs), followed by the plasma membrane and Golgi apparatus (22 SLCs). 124 SLCs (27.7%) had three or more locations annotated, and 29 SLCs (6.5%) had no high-confidence location annotation at all.

## A web portal for integrating data and knowledge on the SLC superfamily

The functional characterization efforts of the RESOLUTE and REsolution consortia created numerous rich and diverse data sets (Fig. 4A). To synthesize the different aspects of SLC function into a comprehensive SLC knowledgebase, we integrated our resources and analyses into a database and consolidated selected information from the public domain, such as subcellular localization, structure, tissue expression, disease associations, availability of assays and chemical compounds as well as knowledge from the literature (Table EV2). We made this SLC-centric data- and knowledgebase available via a web portal (https://re-solute.eu) (Methods; Fig. EV4).

Different sections enable browsing and querying of this unique transporter-focused resource. In addition to sections summarizing

the aims, mission, and project outputs of the RESOLUTE and REsolution consortia, we provide a comprehensive knowledgebase (https://re-solute.eu/knowledgebase) (Fig. 4B). Here, we have integrated general knowledge on SLC genes with transporter-specific annotations and public data sets. A dedicated page for each SLC member covers basic annotations on the gene, transcript and protein levels, along with matched identifiers. It also includes evidence on gene variants, a visualization of the transmembrane topology, subcellular localization and tissue expression levels, the results of the curated literature mining for transported substrates, the results of the data mining for disease associations as well as a summary of available experimental data. In addition to the SLC pages, we have implemented representations of the substrate and disease ontologies, which allow finding and grouping of SLCs by searching or browsing through the ontology graphs of ChEBI and Mondo, respectively.

The resources section (https://re-solute.eu/resources) provides information on and access to reagents, tools, experimental data, and results generated by the consortia. The reagents cover sgRNAs, plasmids and cell lines, and the tools made available to the scientific community are protein binders, functional assays and protocols. Our systematic multi-omics characterization of the generated cell lines led to large-scale experimental data sets, and the key insights from these individual data sets were described in accompanying manuscripts (Wiedmer et al, 2025; Frommelt et al, 2025; Wolf et al, 2025). We present our analyses of these data sets using a range of interactive dashboards to enable the generation of hypotheses on SLC function by visual exploration (Fig. 4C). Custom gene-level annotations can be visualized on the SLC tree in the structural classification dashboard. Dashboards on the transcriptomics and targeted metabolomics data allow exploration of the changed transcripts and metabolites, respectively, in each analyzed cell line (Wiedmer et al, 2025). The changes in measured genes and metabolites across all cell lines can be inspected in dedicated "gene" and "compound" views. The interaction proteomics dashboard enables exploration of bait- or prey-centric interaction networks extracted from the full SLC interactome (Frommelt et al, 2025). To expand the context of a network, external protein-protein interaction resources (currently BioGRID and CORUM are supported) can be overlaid. The genomics dashboard visualizes the integrated genetic interaction network, generated from nine different genetic interaction screens (Wolf et al, 2025). The magnitude and significance of each interaction, as well as the effect of the double knockout compared to the corresponding

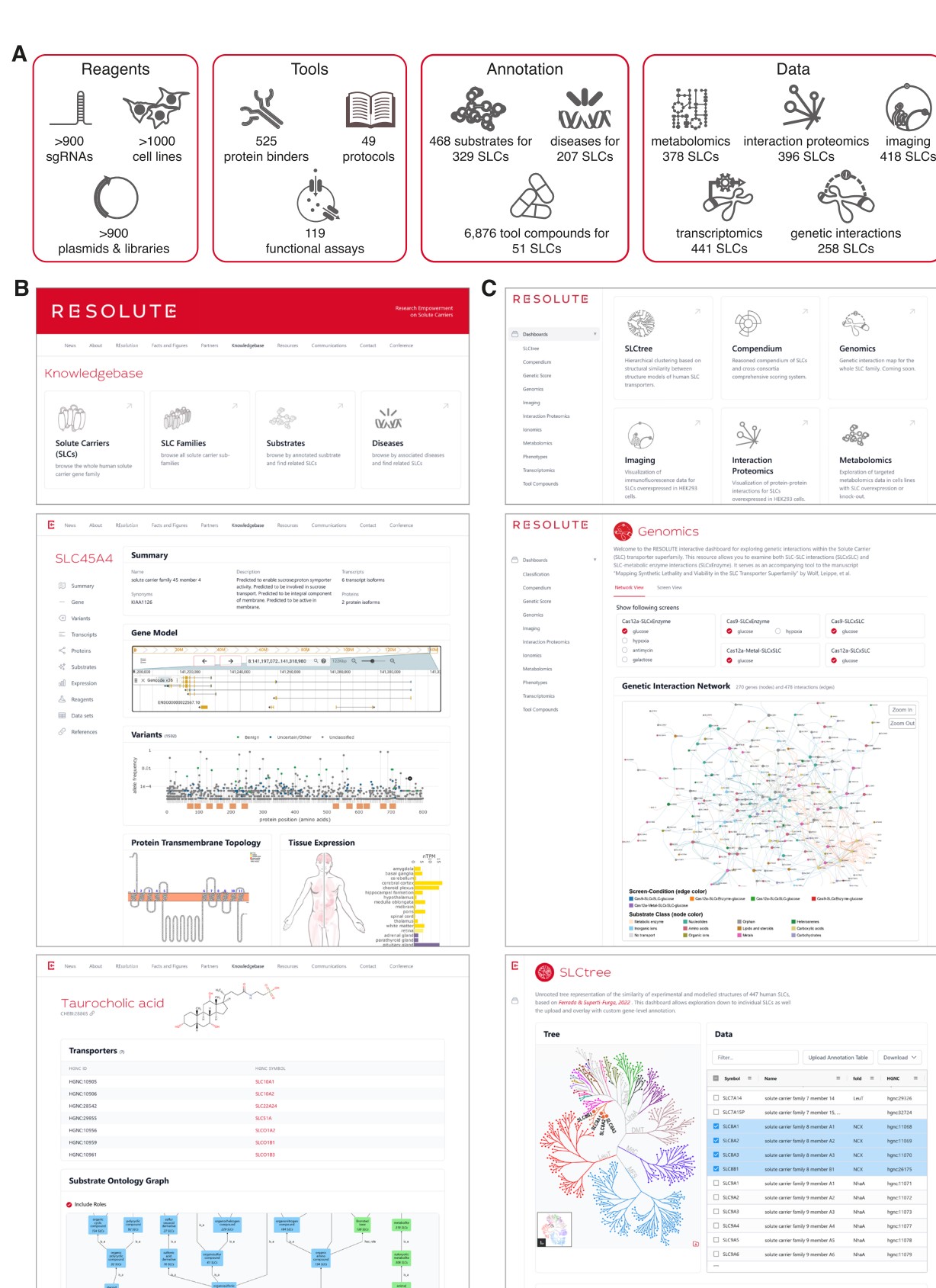

individual single knockouts, can be analyzed at the "screen" view. The imaging dashboard provides access to all immunofluorescence images generated. After selecting a cell line, a stain combination and a view, the corresponding immunofluorescence image can be freely levelled, magnified and downloaded, or compared side-by-side to a different cell line or staining. The tool compounds dashboard shows the results of data mining efforts on active compounds for SLCs (Digles et al, 2024) and can be used to identify potential tool compounds for an SLC of interest. Finally, the compendium dashboard allows an integrated analysis of selected SLC annotations by interactively constructing and navigating a sunburst diagram with an arbitrary number of custom layers and filters. This might lead to insights on interesting relations, e.g., combining fold with subcellular localization and substrate class layers.

To maximize the impact of all generated data, we adhered to the FAIR principles (Wilkinson et al, 2016). Data sets were released as open access using a permissive license. We implemented identifiers and employed standardized, open data formats. Meta data, annotations and analysis results are available as structured data or via an API specifically developed for programmatic access. The raw data sets can be downloaded from the web portal. To ensure sustainability, all data sets are uploaded to public, domain-specific repositories (Goldmann et al, 2025).

## Construction and application of a multimodal functional landscape of the SLC superfamily

To construct a functional landscape of human solute carriers, we integrated eight selected, orthogonal data sets on phenotypic profiles of the SLC superfamily. In addition to the transcriptional and metabolic profiles upon SLC overexpression (Wiedmer et al, 2025), we chose the protein interaction network (Frommelt et al, 2025), subcellular localization annotation, tertiary structure (Ferrada and Superti-Furga, 2022), the tissue expression pattern (Uhlén et al, 2015), substrate annotations as well as disease associations (Table 1). A large majority of SLCs ($n = 404$; 90.4%) were represented in six or more modalities (Fig. EV5A). The selected modalities encompassed different functional aspects and consisted of fundamentally different data types. For successful integration, we transformed measurements or annotations of each modality to a comparable data type and scale, namely SLC–SLC pair similarity matrices. Briefly, we applied data-type-specific distance measures and harmonized the resulting distances to comparable dissimilarities by normalization and transformation (Methods; Fig. EV5B). Correlating the resulting SLC–SLC dissimilarities showed some expected agreements (e.g., structure with substrate annotation, and interaction proteomics with subcellular localization), but the observed correlations were very small (maximum $r_{Pearson} = 0.13$), indicating that each modality provided additional, orthogonal aspects to SLC function (Fig. EV5C). For merging the modality-specific dissimilarities, we employed a weighting scheme based on information content, which was higher for experiment-based dissimilarities than for annotation-based dissimilarities (Fig. EV5D), together with a subjective ranking of functional importance. This resulted in higher weights for transcriptome, metabolome, and interactome profiles, as well as structure. Medium weight was assigned to the substrate annotation and lower weight to subcellular localization annotation, tissue expression, and disease association (Table 1).

Integration resulted in an overall similarity for each of the 99,681 possible SLC–SLC pairs (Fig. EV5E; Dataset EV5). Finally, we employed the UMAP algorithm (McInnes et al, 2018) to represent the data manifold underlying the overall similarity matrix as a graph and derive graph-based distances that capture local as well as global structures of the SLC–SLC similarities. For visualization, the graph was then embedded in a two-dimensional SLC landscape (Fig. 5A). The distances of the eight modalities used as input each showed significant coherence with at least one functional SLC annotation, and the integrated, graph-based distances showed a strong and balanced coherence to all functional SLC annotations, corroborating the validity of the selected modalities and the integration and dimensionality reduction approach (Fig. EV5F).

Clustering the high-dimensional graph underlying the SLC landscape identified 11 distinct communities of SLCs (Fig. 5B). Overall, the clusters recapitulated some of the major folds (Fig. 5C). For example, cluster 8 included all SLCs featuring a LeuT fold, except for the two SLC11 family members that were located in cluster 4 together with other metal transporters. Other folds were spread across two or more clusters, such as the MFS fold, for which 54 and 66 of its 138 members belonged to clusters 1 and 2, respectively (Fig. 5D). The separation of MFS fold members appeared to follow substrate specificity, as 30 out of the 42 MFS fold members with known substrates in cluster 1 were classified as amino acid or carbohydrate transporters, compared to only 3 out of 47 MFS fold members with known substrates in cluster 2. In general, the substrate specificity in cluster 2 appeared to be much more promiscuous.

Cluster 3 provided an example of the contribution of subcellular location to the clustering, resembling the mitochondrial-localized solute carriers (Fig. 5E). While it was dominated by MitC fold members, it also included all sideroflexins featuring the rather distant SLC56 fold. Another interesting member of cluster 3 was SLC30A9, which is a zinc transporter with a YiiP fold. On the SLC landscape, SLC30A9 was relatively dissimilar to the other YiiP fold members, which were all in cluster 4. Indeed, we found SLC30A9 exclusively localized at the mitochondria, supporting previous computational (Kowalczyk et al, 2021) and experimental evidence (Rensvold et al, 2022).

Cluster 4 covered 25 of the 31 metal transporters. Here, substrate class appeared to be the major determinant as the cluster integrated the two well-known, but rather distinct, YiiP and ZIP folds, corresponding to SLC30 and SLC39 family members, respectively (Wiuf et al, 2022; Pasquadibisceglie et al, 2022; Zhang et al, 2023). Next to those two typical families for metal transport, the landscape also positioned SLC11A2—also known as Divalent Metal Transporter 1 (DMT-1)—at the center of this cluster (Fig. 5F). In particular, the manganese transporters SLC39A8 and SLC39A14 were two of the four closest SLCs to SLC11A2. Inspecting the similarities on each of the eight modalities that were used to construct the landscape revealed high similarities in targeted metabolome profiles, interaction networks, annotated substrates and associated diseases (Fig. 5G). Indeed, previous studies have shown that all three proteins can transport the same divalent cations (Polesel et al, 2023; Garrick et al, 2006). In addition, manganese and iron homeostasis is dysregulated in various animal models when transporter function is disrupted (Fujishiro et al, 2012; Tuschl et al, 2013; Chen et al, 2018). There is also evidence that altering SLC39A14 expression levels influences the expression of

**Table 1.    Data sets integrated in the functional SLC landscape.**

| Dataset | Source | SLCs covered | Data type | Distance measure | Integration weight |
|---|---|---|---|---|---|
| Transcriptomics | RESOLUTE (Wiedmer et al, 2025) | 441 | Profiles of differential transcript abundances | Euclidean | 3 |
| Targeted metabolomics | RESOLUTE (Wiedmer et al, 2025) | 378 | Profiles of differential metabolite abundances | Euclidean | 3 |
| Interaction proteomics | RESOLUTE (Frommelt et al, 2025) | 396 | Sets of interacting proteins | Co-occurrence of interactors | 3 |
| Subcellular localization | RESOLUTE (this study) | 418 | Annotated lists of cellular compartments | Term co-occurrence | 1 |
| Structure | Ferrada and Superti-Furga, 2022 | 447 | Atomic coordinates | Structural alignments | 3 |
| Tissue expression | Human Protein Atlas (Uhlén et al, 2015) | 443 | Profiles of gene expression in tissues | Correlation-based | 1 |
| Substrate annotation | This study | 329 | Annotated lists of chemical compounds | Chemical fingerprints | 2 |
| Disease association | REsolution (this study) | 226 | Annotated lists of disease associations | Semantic similarity | 1 |

SLC39A8 and SLC11A2 in different mouse organs, further suggesting that these transporters are functionally related (Xin et al, 2017). Together, these findings supported the remarkably close positioning of these three transporters on the functional SLC landscape, regardless of their different structural folds.

The quality and coherence of the clusters were assessed via their mean silhouette scores (Rousseeuw, 1987), which were positive for all clusters and ranged from 0.08 for the more loosely defined cluster 1 up to 0.73 for the very coherent cluster 11 (Fig. EV5G). In a mutual information analysis, we systematically compared the clusters on the landscape with a defined set of discrete SLC properties (Fig. EV5H). While the fold annotation showed the highest coherence with the landscape clusters, family grouping and substrate classification also had considerable mutual information with the overall clustering. The number of available modalities shared no mutual information with the clustering, suggesting that the clustering was not driven by the incomplete coverage of the data sets. A detailed enrichment analysis found between 2 and 9 significantly overrepresented properties of SLCs in each cluster, and 79 properties enriched overall, demonstrating that the integrative clustering approach was effectively capturing actual functional similarity (Fig. EV5I; Dataset EV6). In a cross-validation study, omitting any of the modalities for building the SLC landscape reduced the number of enriched properties, underlining that each of the modalities chosen contributed relevant information to the functional SLC landscape (Fig. EV5J).

In summary, the SLC functional landscape provides a useful, integrative, and comparative representation of the properties of this rather understudied superfamily of membrane transporters and, as a tool, offers insights beyond those that can be gained by manual inspection of the different data sets and annotations for individual transporters.

# Discussion

The complexity of the task of assigning functional properties to a large group of gene products that are related only by a shared principle, that of transporting inorganic and organic molecules across lipid bilayers without hydrolyzing ATP, is enormous (César-Razquin et al, 2015; Superti-Furga et al, 2020; Hediger et al, 2004). Human membrane transporters seldom transport only one chemotype, and functional redundancy is built-in into circuits fundamental for all cellular functions, such as the provision of energy and macromolecular building blocks, control of ionic strength and osmolarity for protein folding and chemical reactions, control of pH, redox and membrane potential. In retrospect, it is difficult to think of another class of proteins whose function is as broadly intertwined with basic biophysical effects as SLC transporters. In comparison, GPCRs, transcription factors, proteases, and RNA-binding proteins all may have very large individual effects but are all essentially conveying signals. Transporter activity, however, changes the very chemical makeup of the environment the proteins operate in. Thus, direct functional consequences in the systematic interrogation of SLCs are obscured by a plethora of compensatory mechanisms and hinder the unambiguous assignment of transporter substrates and functions.

Despite these inherent difficulties, we have been able to assign functional properties to each SLC, predominantly defined by the integration of measurements upon elevated protein expression. In total, we used dimensions derived from eight different modalities to calculate an integrated virtual landscape in which each SLC finds a specific location based on its corresponding data and annotation. Although this may not always allow derivation of a definitive or unequivocal functional assignment to each SLC superfamily member, it allows for a meaningful comparison of neighborhoods and relative multidimensional distances of properties between SLCs. The molecular profiles employed in this study (transcripts, metabolites, interacting proteins), representing measurements derived from the genetic control of protein expression under comparable, controlled conditions, were weighed more strongly when deriving the functional landscape. We think that this is where the power of the approach lies, which became only possible with the advent of CRISPR/Cas technology (Bock et al, 2022; Wang and Doudna, 2023). It is common to find sceptics of large-scale biology arguing that only in-depth dedicated studies allow for real new biological insights. While parallel approaches cannot replace dedicated studies, it is precisely the comparative nature of parallel assessment under controlled conditions that allows for objective

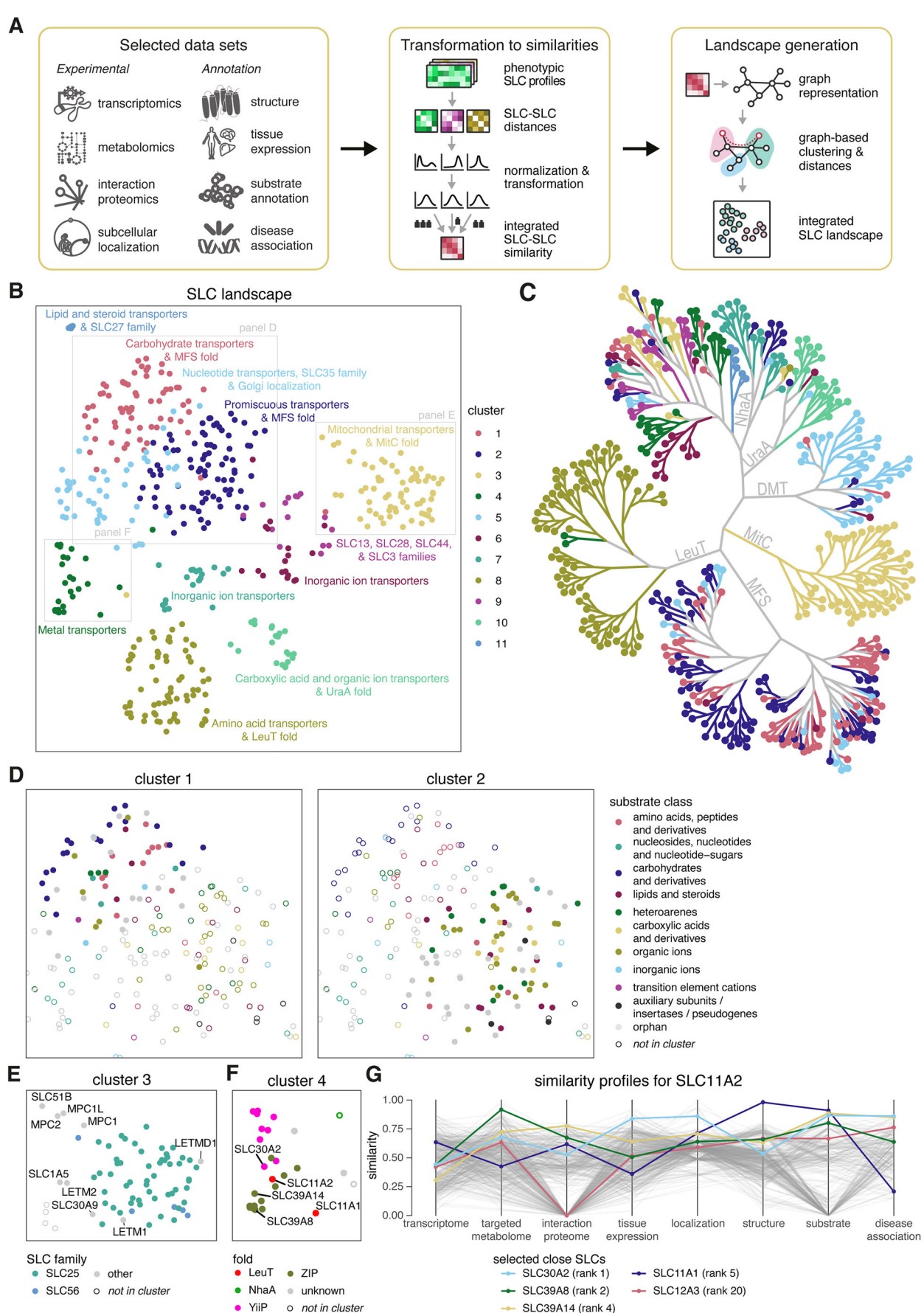

◄

**Figure 5. An integrated, functional landscape of the human SLC superfamily.**

(**A**) Schematic overview of the workflow generating the SLC landscape. The eight selected source data sets are illustrated in the first box. The second box details the transformation of source data to integrated SLC–SLC similarities, and the third box indicates how the UMAP algorithm was used to derive a neighborhood graph and the final landscape. (**B**) The two-dimensional representation of the functional landscape of 447 SLCs. Results of the high-dimensional graph-based clustering are indicated by color and labeled by simplified but expressive cluster names. Selected areas from (**D–F**) are indicated by gray boxes. (**C**) The cluster membership of each SLC is visualized on the structure-based SLC tree, using the same color scheme as in (**B**). The six largest folds are labeled on their respective branch. (**D**) Cut-out of the functional SLC landscape highlighting clusters 1 and 2 and demonstrating the different predominance of substrate classes. (**E**) Cut-out of the functional SLC landscape highlighting cluster 3 and demonstrating the overrepresentation of the mitochondrial SLC25 family members. The mitochondrial zinc transporter SLC30A9 is highlighted. (**F**) Cut-out of the functional SLC landscape highlighting cluster 4 and demonstrating the two major folds of metal transporters (YiiP and ZIP). SLC11A2, featuring a diverging LeuT fold, and some of its neighbors are highlighted. (**G**) Profile plots of the similarity of all SLCs to SLC11A2, in each of the eight modalities that were used in constructing the functional landscape. Selected SLCs, which are close on the landscape, are highlighted, among them the two manganese transporters SLC39A8 and SLC39A14.

conclusions (Hieter and Boguski, 1997; Ideker et al, 2001; Collins et al, 2003). Until now, most insights on SLC protein function addressed only one or a few members of an SLC family or of SLCs acting in concerted action. It is the global, integrative and comparative nature of the SLC landscape that allows for the identification of accrued properties not inferable by subfamily membership alone.

The value of the experimental data obtained in this concerted effort is potentially limited by several factors mentioned in the individual accompanying manuscripts (Wiedmer et al, 2025; Frommelt et al, 2025; Wolf et al, 2025). One limitation that is apparent also in the experimental data set for subcellular localization described here is that protein overexpression models may lead to protein accumulation in the ER and therefore an overrepresentation of annotations to this location or connected compartments such as the nucleus and Golgi. This is reflected in the proportionally large number of novel assignments to these compartments by the RESOLUTE data set. However, for systematic experimental subcellular localization, it is necessary to use over-expression models due to the lack of specific antibodies, particularly for SLCs (Gelová et al, 2024; Ayoubi et al, 2023). This limitation is also apparent from the observation that HPA, consisting of antibody-based evidence of subcellular localization, showed the least agreement with other annotation sources. Importantly, the SLC superfamily-wide localization data presented in this study provides the first experimental evidence for 48 SLCs. Overall, it is a valuable resource for understanding SLC function as well as for assay design and drug discovery, for which knowledge of subcellular localization is crucial (Dvorak et al, 2021).

To partly correct for specific biases of the experimental approaches, we include data modalities that were unlikely affected by similar biases, such as protein structure analysis, chemical substances known to be transported by individual SLCs, and genetic association with diseases. While the protein structure analysis using AlphaFold2 predictions was published recently (Ferrada and Superti-Furga, 2022), the substrate annotation and disease associations required systematic collection, annotation and curation. Compared to the initial annotation of SLC substrates a few years back (Meixner et al, 2020), our integrative analysis required stronger experimental evidence for substrates. Despite the more stringent criteria applied, the fact that research on transporters was gaining traction over the last years led to a higher number of SLCs with known substrates. The annotation comprises a twofold redundancy: a substrate can be transported by different SLCs, and a transporter can have numerous substrates. In the future, such information may be used in rationalizing the choice of

drug targets, acknowledging that some SLCs may have more diffuse effects due to their relative promiscuity. However, substrate annotation is far from complete. Discovery of the full panel of substrates might be partially hindered by the common use of gene/protein names that imply a particular function as a given fact, thereby demotivating research efforts on more physiologically relevant functions or yet unknown substrates. For example, SLC2A1 (currently named "solute carrier family 2 member 1") had a previous name "human T-cell leukemia virus (I and II) receptor". In this sense, the use of the HUGO nomenclature for SLC genes and proteins leads to less bias than the one induced by historically adopted, imprecise names. Moreover, the limitation of any literature-based annotation remains the collective bias of research topics towards well-studied areas, resulting in under-studied genes (Edwards et al, 2011).

A comprehensive overview of pathogenic genetic alterations in SLC genes has not been available since genetic studies are typically focused on individual SLCs and specific diseases (Li et al, 2021), families (Kölz et al, 2021), refer only to a limited population (Rajman et al, 2020; Mir et al, 2022), or contain only limited genetically-encoded aspects (Schaller and Lauschke, 2019). We undertook a systematic data mining and curation process for all SLCs, consolidating previously fragmented information. The curated data set represents a unique resource of genetic, clinical, and experimental evidence, and enabled the inclusion of under-studied SLCs, which have not yet been clinically recognized, revealing translational implications for mutations observed in different SLC families and individual SLCs. One limitation of the presented human genetics data set is potential biases, such as population-based variations and sample size constraints inherent in the mined databases (Fitipaldi and Franks, 2023). These biases may skew the representation of certain genetic variants, particularly when certain populations are not well represented (Schaller and Lauschke, 2019), making it difficult to generalize findings across diverse populations. Another challenge arises from the nature of GWAS, where individual associations may result in false positives due to linkage disequilibrium (Slatkin, 2008).

The data- and annotation-informed integration into a global SLC landscape allowed us to define functional distances between SLCs and group them into 11 clusters, each resembling a mix of different known properties of SLCs. In addition to investigating an SLC of interest via the transporters featuring the most similar phenotypic profiles, the integrated landscape allowed the extension of this guilt-by-association principle to the functionally closest genes, combining local and global features of the integrated data sets. Examining the landscape enabled us to uncover interesting

relations between SLC family members. We found the iron transporter SLC11A2 located close to the manganese transporters SLC39A8 and SLC39A14. The similarity between these three transporters could be beneficial for designing novel therapeutics. Loss-of-function mutations in SLC39A8, SLC39A14, or SLC11A2 lead to three distinct clinical presentations. Bi-allelic mutations in SLC39A8 or SLC39A14 lead to the rare diseases Disorder of Glycosylation type IIn or Hypermanganesaemia with Dystonia 2, respectively, both characterized by abnormal blood Mn levels (Tuschl et al, 2013; Winslow et al, 2020; Anagianni and Tuschl, 2019). Conversely, SLC11A2 loss of function or reduced expression results in microcytic anemia (Mims et al, 2005; Beaumont et al, 2006; Shawki et al, 2015). The different clinical phenotypes may result from differences in Mn or Fe binding affinity or total transporter abundance in vivo; therefore, altering the abundance of a similar transporter may restore metal homeostasis. For example, pharmacologically increasing gene expression of SLC39A8 in the duodenums of patients encoding SLC11A2 loss-of-function mutations could increase Fe levels in this organ, since both proteins mediate metal transport across apical membranes of polarized cells (Gunshin et al, 1997; He et al, 2006), and SLC39A8 expression leads to increased iron uptake in HEK293 cells (Wang et al, 2012).

Overall, this example demonstrates that the integrated similarity between transporters can be used to generate therapeutic strategies for SLC-related disorders.

Beyond the few examples highlighted in this study, the landscape offers much more to explore. The functional allocation is only a starting point, providing the groundwork for real functional implications of hundreds of SLCs that were uncharted before and multiplying the previously collected knowledge on the subject. A limitation due to the simplification of a two-dimensional representation of the landscape is that the closeness between two transporters is not always reflected in the vicinity on the depicted map of the landscape, and visual interpretability may profit from novel visualization techniques (Hütter et al, 2022). Variations of the landscape with a different functional focus could be generated by tuning the weights of the integrated data sets or by the inclusion of additional dimensions. Transforming the selected data sets to SLC–SLC similarities allowed for more straightforward integration but also sacrificed potentially valuable information. Integration via prior-knowledge networks could lead to successful integration while preserving most information, and at the same time open up interesting applications of machine learning (Fortelny and Bock, 2020). While not used in this study, prior-knowledge networks have

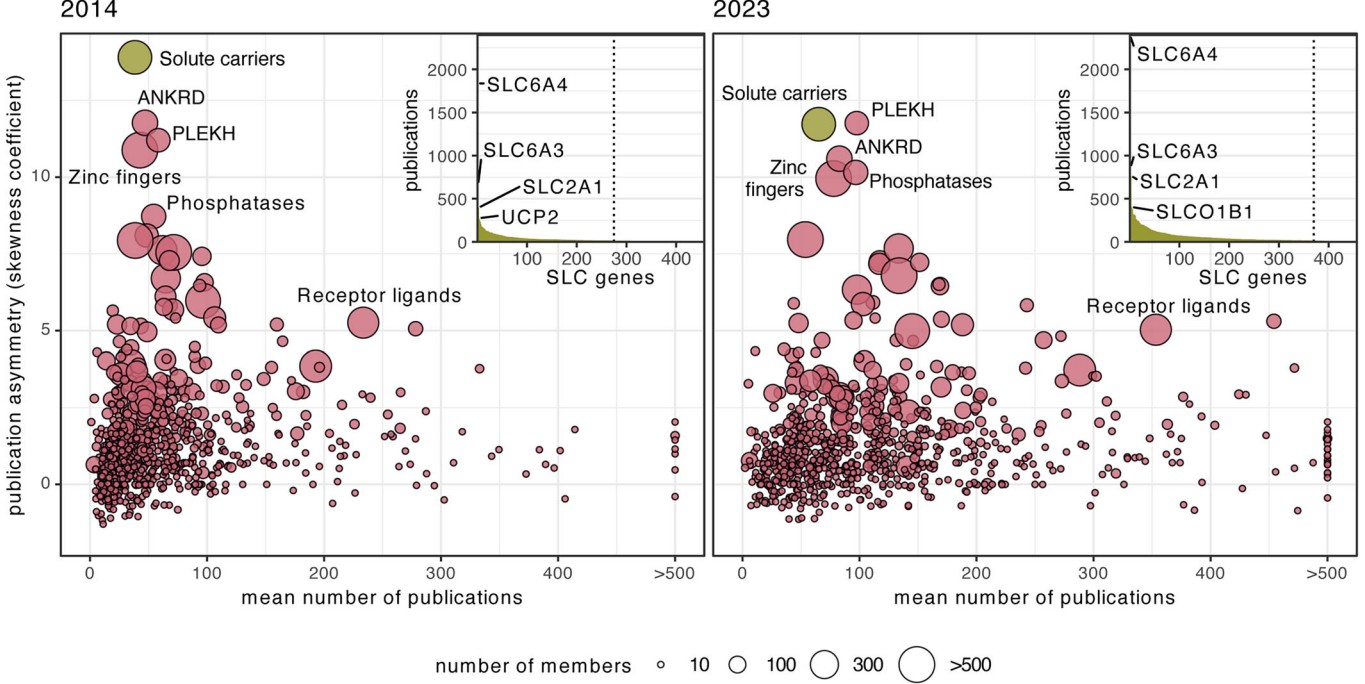

**Figure 6. Evolution of publication asymmetry over the last 10 years.**

Publication asymmetry of gene families compared to their members' mean number of publications, as shown in (César-Razquin et al, 2015). Publications were considered up to the end of 2014 on the left panel and up to the end of 2023 on the right panel. Over the last decade, publication asymmetry increased for some families, such as the group of pleckstrin homology domain-containing (PLEKH) genes or the group of phosphatases, it remained constant for families such as the group of receptor ligands, and decreased for other families, particularly for the solute carrier superfamily. Point size corresponds to the size of the gene family, and selected gene families are labeled. The inset ranks the SLC genes by their number of associated publications, demonstrating the asymmetry within this superfamily. The four most studied SLCs are indicated. The dashed line marks the threshold to SLCs with fewer than 15 associated publications, corresponding to 181 SLCs as of 2014 and 86 SLCs as of 2023. Publication counts per gene are based on the "gene2pubmed" and "generifs_basic" files provided and curated by NCBI. Gene groups as defined by HGNC. Only protein-coding genes were considered. Asymmetry is measured for each group of genes by calculating the skewness ("moments" library, version 0.14.1) of the distribution of the number of publications for all genes within the group (Dataset EV7). A high skewness coefficient indicates a distribution where a few genes in the family concentrate a much higher number of publications than the rest.

been used at a smaller scale by employing the Reactome pathway database for linking gene expression to metabolite abundances (Wiedmer et al, 2025; Milacic et al, 2024). A promising route will be the integration of our comprehensive data sets on human solute carriers to knowledge graph initiatives, such as the recently published BioCypher (Lobentanzer et al, 2023). New studies discovering novel functional aspects of individual SLCs will allow validation of the effectiveness of the presented integration approach, retrospectively.

When we confronted the community with the necessity to mount a concerted action on human solute carriers ten years ago, the knowledge landscape was different (César-Razquin et al, 2015). Less than a handful of experimentally determined structures of human SLCs were available, and the main functional assay consisted of injecting human mRNA encoding an SLC into frog oocytes and measuring the differential accumulation of radio-metrically labeled suspected substrates. Given the importance of the interface between the chemical and biochemical worlds, and the fact that it is mainly governed by transporters, a call for action seemed appropriate. Where do we stand ten years later? The research landscape on SLCs has changed dramatically, such that the SLC family recently lost its title of the "most asymmetrically studied gene family" to the family of pleckstrin homology domain-containing proteins (Fig. 6). There is a new level of awareness and engagement in SLC research, as exemplified in some spectacular recent studies (Wang et al, 2021; Morioka et al, 2018; Luongo et al, 2020; Adelmann et al, 2020; Liu et al, 2023). The ability to chemically address the target class has improved dramatically (Wang et al, 2020; Galetin et al, 2024; Dvorak and Superti-Furga, 2023), and there has been an explosion of three-dimensional structures elucidated via cryo-EM that has not only clarified transport mechanisms but also revealed the mechanism of action of important therapeutics (Parker et al, 2021; Parker and Newstead, 2017; Coleman et al, 2019; Côté et al, 2011; Srivastava et al, 2024). Moreover, a wider role of SLCs in drug uptake has been recognized (Girardi et al, 2020). There are many more functional assays than a decade ago, especially involving intact human cells.

To empower future research on solute carriers, we set up a web portal allowing the scientific community to access and explore the resources, tools, analyses, and data sets collected and generated by the RESOLUTE and REsolution consortia. This knowledgebase, together with the newly generated toolbox of assays (Dvorak et al, 2021; Digles et al, 2024), protein binders (Gelová et al, 2024), plasmids and cell lines (Wiedmer et al, 2025), and omics data sets (Wiedmer et al, 2025; Frommelt et al, 2025; Wolf et al, 2025), should represent a turning point in the ability of the community to work on this superfamily of proteins. We anticipate the continued decrease of the publication asymmetry of SLCs, manifested by an acceleration in the process of creating a more even engagement of the research community with this superfamily. The threshold should be lowered as no SLC is now likely to be completely void of functional suggestions. We are convinced that the substantial effort that was mounted in the last years represents a formidable starting point for research on the largest group of membrane transporters encoded in the human genome and a potential blueprint for similar efforts on other groups of neglected human gene products.

# Methods

## Reagents and tools table

| Reagent/resource | Reference or source | Identifier or catalog number |
|---|---|---|
| **Experimental models** | | |
| Jump-In™ T-REx™ HEK293 cells | Thermo Fisher Scientific | RRID:CVCL_YL74 |
| **Recombinant DNA** | | |
| **Antibodies** | | |
| Anti-CD107b (LAMP2) mAb (H4B4), AlexaFluor488 | Thermo Fisher Scientific | #53-1078-42 |
| Mouse anti-PMP70 | Sigma | #SAB42000181 |
| Mouse anti-KDEL MAB (10C3) | Enzolifesciences | #ADI-SPA-827-F |
| Mouse anti-Tubulin (D3U1W) | Cell Signalling Technology | #86298S |
| Rabbit anti-GOLGB1/Giantin | Novus | #NBP1-91938 |
| Rabbit anti-RAB9A (SR45-05) | Novus | #NBP2-67331 |
| Rabbit anti-NaK ATPase | Abcam | #76020 |
| Goat anti-HA | Novus | #NB600-362 |
| Mouse anti-HA | Thermo Fisher Scientific | #26183-1MG |
| Donkey anti-Mouse IgG (H + L), Alexa Fluor Plus 488 | Thermo Fisher Scientific | #A32766-1MG |
| Donkey anti-Rabbit IgG (H + L), Alexa Fluor Plus 488 | Thermo Fisher Scientific | #A32790-1MG |
| Donkey anti-Rabbit IgG (H + L), Alexa Fluor Plus 555 | Thermo Fisher Scientific | #A32794-1MG |
| Donkey anti-Mouse IgG (H + L), Alexa Fluor Plus 555 | Thermo Fisher Scientific | #A32773-1MG |
| Donkey anti-Goat IgG (H + L), Alexa Fluor Plus 647 | Thermo Fisher Scientific | #A32849-1MG |
| **Oligonucleotides and other sequence-based reagents** | | |
| **Chemicals, enzymes, and other reagents** | | |
| DPBS+Ca+Mg | Thermo Fisher Scientific | 14040133 |
| DMEM | Thermo Fisher Scientific | #31966 |
| FCS h.i., dialyzed | Amimed | #2-01F17-I |
| NEAA (100x) | Thermo Fisher Scientific | #11140-035 |
| Hepes (1M) | Thermo Fisher Scientific | #15630-056 |
| Penicillin / Streptomycin (PS) (100x) | Thermo Fisher Scientific | #15140-122 |
| Digitonin | Thermo Fisher Scientific | #407565000 |
| Triton-X100 | Thermo Fisher Scientific | #85111 |
| Paraformaldehyde 32% w/v | Electron Microscopy Sciences | 15714-S |
| Doxycyclin-hydrochloride | Sigma-Aldrich | D3447-500MG |
| Hoechst 33342 | Thermo Fisher Scientific | #H3570 |
| HCS Cellmask DeepRed | Thermo Fisher Scientific | #32721 |

| Reagent/resource | Reference or source | Identifier or catalog number |
|---|---|---|
| Mitotracker Orange CMTMRos | Thermo Fisher Scientific | #M7510 |
| Laminin 5-2-1 | Biolamina | LN521 |
| **Software** | | |
| R Project for Statistical Computing v4.3.3 | https://cran.r-project.org | RRID:SCR_001905 |
| 'bestNormalize' R library v1.9.1 | CRAN (The Comprehensive R Archive Network) | |
| 'bigdatadist' R library v1.1 | CRAN (The Comprehensive R Archive Network) | |
| 'ChemmineR' R library v3.54.0 | Bioconductor | |
| 'ggraph' R library v2.2.1 | CRAN (The Comprehensive R Archive Network) | RRID:SCR_021239 |
| 'leiden' R library v0.4.3.1 | CRAN (The Comprehensive R Archive Network) | |
| 'moments' R library v0.14.1 | CRAN (The Comprehensive R Archive Network) | |
| 'simona' R library v1.0.10 | Bioconductor | |
| 'truncnorm' R library v1.0-9 | CRAN (The Comprehensive R Archive Network) | |
| 'vegan' R library v2.6-4 | CRAN (The Comprehensive R Archive Network) | |
| Python Programming Language v3.11.6 | https://www.python.org | RRID:SCR_008394 |
| 'leidenalg' Python library v0.10.1 | PyPI (The Python Package Index) | |
| 'sklearn' Python library v1.3.2 | PyPI (The Python Package Index) | |
| 'umap-learn' Python library v0.5.3 | PyPI (The Python Package Index) | |
| Node.js v20 | https://nodejs.org | |
| AG Grid v26.1.0 | npm (Node package manager) | |
| Plotly v2.27.0 | npm (Node package manager) | RRID:SCR_013991 |
| sigma.js v3.0.0 | npm (Node package manager) | |
| Tailwind CSS v3.3.2 | npm (Node package manager) | |
| urql v3.0.3 | npm (Node package manager) | |
| Docker v24.0.2 | https://www.docker.com | |
| Express.js v4.17.1 | https://expressjs.com | |
| Gephi v0.10.1 | https://gephi.org | RRID:SCR_004293 |
| GitLab v14.9.0 | https://gitlab.com | RRID:SCR_013983 |
| nginx v1.24.0 | https://nginx.org | |

| Reagent/resource | Reference or source | Identifier or catalog number |
|---|---|---|
| PostGraphile v4.13.0 | https://postgraphile.org | |
| PostgreSQL v15 | https://www.postgresql.org | RRID:SCR_021067 |
| Prisma v5.5.2 | https://www.prisma.io | |
| React v18.2.0 | https://react.dev | |
| **Other** | | |
| HCI 384 well microplates | Costar | #4518 COC |
| Yokogawa CV7000 | Yokogawa Instruments, Japan | |

All analyses were conducted in R (version 4.3.3), unless stated otherwise. No blinding was performed in this study.

## Constructing the SLC tree

Pairwise structural similarities were obtained from (Ferrada and Superti-Furga, 2022). Distances were calculated by subtracting similarities from 1 and used for hierarchical clustering using Ward's criterion (Ward, 1963). Branch lengths of the resulting dendrogram were square-root scaled and visualized as an unrooted tree using the equal daylight algorithm of the "ggraph" library (version 2.2.1). The tree was further semi-automatically adjusted using Gephi software (version 0.10.1) (Bastian et al, 2009).

## Substrate annotation, ontology mapping, and substrate class definitions

A team of 16 people in the Superti-Furga group (graduate students and postdoctoral fellows) scanned the primary literature on human solute carriers. The criteria for annotating a compound as a substrate were more strict than in our previous annotation effort (Meixner et al, 2020). Only reports with proof from a transport assay using human solute carrier proteins were considered; no homology-based inferences were allowed. Collected compound names were manually mapped to ChEBI terms (Hastings et al, 2016). Biologically corresponding terms (tautomers, conjugated bases/acids) were identified and harmonized. For mapped generic terms, which describe not a single compound but rather a class of compounds in ChEBI, the primary literature was re-checked to find and match a more specific chemical compound. After a number of iterations, we arrived at 2044 SLC-compound-publication pairs. Every SLC substrate annotation has at least one reference to a primary resource, mainly via PubMed identifier.

We defined a set of nine biologically and biochemically relevant substrate classes based on manual selection of higher-level ChEBI terms (listed in parentheses): "lipids and steroids" (CHEBI:18059, CHEBI:35341), "nucleosides, nucleotides and nucleotide-sugars" (CHEBI:33838, CHEBI:36976, CHEBI:25609), "amino acids, peptides and derivatives" (CHEBI:33709, CHEBI:16670, CHEBI:22860, CHEBI:37793, CHEBI:63534), "carbohydrates and derivatives" (CHEBI:78616, CHEBI:24848), "heteroarenes" (CHEBI:33833, CHEBI:17015), "carboxylic acids and derivatives" (CHEBI:33575, CHEBI:37622), "transition element cations" (CHEBI:33515), "organic ions" (CHEBI:25699), "inorganic ions" (CHEBI:36914). Annotated

substrate terms were then matched to those terms, walking down the ontology tree on "is a", "is conjugate base of", "is conjugate acid of", "is tautomer of", and "has role" relations. Then we summarized the classification of substrate terms per SLC. Here, an SLC often matched to different classes, either due to multiple annotated substrates matching to different classes, or also due to a single annotated substrate matching to multiple classes. Cases of ambiguous class matching were automatically resolved using the first match based on the order of the classes listed above, which starts with more specific classes and ends with more generic classes. Some SLC classifications were manually curated afterward to better represent the majority of substrates or to better adhere to a common family substrate class.

## Disease association mapping

For all variants collected, we assessed the consequences on the canonical transcripts of all SLC genes as annotated in gnomAD via Sequence Ontology (version 2.5.3) (Eilbeck et al, 2005) terms. Variants were then filtered for consequences from the "protein_altering_variant" (SO:0001818) sub-ontology graph.

Next, collected traits were mapped to selected ontologies: Mondo (Vasilevsky et al, 2022), EFO (Malone et al, 2010), HPO (Köhler et al, 2021), or GO (Ashburner et al, 2000). The mapping workflow relied on three publicly available resources. The first was the Python module OnToma, which processed identifiers from other ontologies and also free text labels to return a corresponding EFO term. For identifiers, we considered the first-ranked successful match. For free text labels, the process included searching for an exact name match from the EFO OT slim OWL file, an exact synonym match from the OWL file, a mapping from a manual string-to-ontology database, and a high-confidence mapping from EBI's ZOOMA tool with default parameters. ZOOMA leverages its manually curated annotations from publicly available databases to find potential matches, accompanied by confidence scores. For traits that remained unmapped by OnToma, ZOOMA was used with more lenient confidence thresholds as the second step of the mapping process. The outcomes were manually reviewed to eliminate obvious inaccuracies. Traits that were still unmapped after this secondary step were considered not related to biological associations. The final step involved using EBI's OxO (Ontology Cross-Reference Service), developed to facilitate the identification of cross-references and synonyms across different ontologies, vocabularies, and coding standards. This allowed the separation between "pathogenic" traits, associated with terms from Mondo, and "non-pathogenic" traits, associated with terms from either EFO, HPO, or GO. There were three rather unspecific Mondo terms, which we manually excluded from the "pathogenic" set: "alcohol-induced mental disorder" (MONDO:0002326), "generalized anxiety disorder" (MONDO:0001942), and "Mendelian disease" (MONDO:0003847).

Disease areas were defined as the 37 direct child terms of the "human disease" term in Mondo (MONDO:0700096). A disease association was counted for a specific disease area if it mapped to the disease term or any more specific term in the ontology. Due to the ontology structure, a single disease term may be categorized under multiple disease areas.

## Immunofluorescence imaging

Jump-In T-REx HEK293 cells were generated, cultured, and quality controlled as described by (Klimek et al, 2022; Wolf and Seuwen, 2021). All cell lines underwent routine testing to confirm the absence of mycoplasma contamination, and authentication of cell lines was performed via western blotting and RNA sequencing. The detailed protocol for immunofluorescence imaging was described by (Pfeifer et al, 2022). In brief, cells were seeded on laminin-coated COC high-content imaging plates at seeding densities adjusted to yield 80% confluency at the beginning of the staining procedure. After 2 days for attachment and proliferation, cells were treated with 1 µg/ml doxycycline to induce transgene expression for 22 h or were left untreated. Doxycycline was washed out and cells were grown for another 18 h. Antibodies or chemical probes were applied to stain 9 cellular compartments and the SLC-HA-tag either prior to or post-fixation and permeabilization according to the manufacturer's recommendations. Cells were incubated with Mitotracker Orange CMTMRos for 30 min, prior to fixation with 3.5% PFA for 25 min and washing. Afterwards, samples were permeabilized and blocked with Triton-X, Digitonin and FBS for 30 min, followed by washing. Samples were incubated with primary antibodies for 2 h at 37 °C (Hoechst 33342, HCS Cellmask DeepRed, anti-KDEL, anti-Giantin, anti-LAMP2, anti-PMP70, anti-RAB9A, anti-Na-K-ATPase, anti-Tubulin, anti-HA-tag), washed, incubated with fluorophore-labeled secondary antibodies for 90 min at 37 °C and washed again. Plates were imaged with the Yokogawa CV7000 using a ×60 water immersion objective. Images were assessed visually by scaling manually to comparable intensities and attributing scores according to the intensity per individual compartment over the total intensity and assigning a confidence score from 1 to 5, with 5 as the highest confidence level. In the process of creating a machine learning-based prediction tool for the localization of a given protein in HEK293 cells (Baranowski et al, in preparation), repeated training iterations of machine learning and the study of mismatches between human annotation and prediction led to several revisions and a final, thoroughly curated SLC localization annotation list.

## Mapping of subcellular localization annotations

Immunocytochemistry-based annotation of subcellular localizations were obtained from the Human Protein Atlas (HPA; version 23.0) (Thul et al, 2017). From the UniProt database (version 2024_01), we obtained Gene Ontology (GO) based subcellular location annotations and the curated set of UniProt annotations. Based on the manual annotation of the RESOLUTE immunofluorescence data set, we selected 10 matching subcellular location terms from the GO and identified matching terms in HPA and UniProt. The selected terms were automatically integrated with annotations for any of their subterms, resulting in consideration of 1823 terms in GO, 158 terms in UniProt, and 22 terms in HPA. There was no matching term for cell projection in HPA. We ranked the reliability scores and evidence codes provided by the different annotation sources and considered only the annotation with the highest score in cases of multiple annotations for an SLC to a selected location term within the same annotation source (Fig. EV3A). We then applied a strict filtering to define the high-confidence subcellular

localization annotations: for RESOLUTE annotations the confidence score had to be 5 and the signal proportion had to be at least 0.2; UniProt annotations had to have the evidence tag "experimental evidence used in manual assertion" (ECO:0000269); GO annotations had to have the evidence code "inferred from direct assay (IDA)", "inferred from mutant phenotype (IMP)", or "inferred from high-throughput direct assay (HDA)"; and HPA annotations had to have a reliability score of "Enhanced", "Supported", or "Approved" (Fig. 3C).

## Implementation of an SLC-centric web portal

The individual components of the web portal (Fig. EV4) were developed in a GitLab environment (version 14.9.0) and deployed in containers (Docker version 24.0.2).

A relational database in a PostgreSQL system (version 15) served as the data layer for the web portal. To populate the database, we employed TypeScript or Python to develop ETL (extract, transform, load) pipelines for processing result tables of our omics data analyses or public data. Pipelines integrating publicly available data were operating directly on APIs or on downloadable snapshot or source files. Prisma (version 5.5.2) served as an object-relational mapper, streamlining the implementation between the data layer and ETL pipelines.

The web server was implemented in Node.js (version 20) using the Express framework (version 4.17.1). An application programming interface (API) was implemented via PostGraphile (version 4.13.0), providing machine-readable access to the data using the GraphQL query language (https://re-solute.eu/graphiql).

The frontend was implemented in TypeScript using the component-based framework React (version 18.2.0) and was served by an nginx server (version 1.24.0). The GraphQL client urql (version 3.0.3) was handling communication to the backend. Several libraries were employed to implement the user interface and enhance the user experience, in particular Plotly (version 2.27.0) for interactive plots and charts, sigma.js (version 3.0.0) for interactive graphs and networks, AG Grid (version 26.1.0) for customizable data grids, and Tailwind (version 3.3.2) for general styling of components.

Not all data was available in the relational database and therefore accessible via the API. Some data sets were served from the web server in structured data files (JSON format), while all larger data sets (in particular, omics raw files) were available from the Microsoft Azure cloud storage.

## Calculating an SLC–SLC similarity matrix

Integration of individual data modalities was achieved by transforming each data set to a similarity matrix. For each modality, we first calculated pairwise distances per SLC–SLC combination (Fig. EV5B). For transcriptomics and metabolomics, the distances corresponded to the Euclidean distance between standard-normalized profiles of logarithmized fold-changes of measured transcripts and metabolites upon SLC overexpression, respectively (Wiedmer et al, 2025). For interaction proteomics, the distance was the Jaccard distance between the sets of identified prey proteins, as detailed in (Frommelt et al, 2025). For tissue expression, the distance was based on the Spearman correlation ($d = \sqrt{(1 - r_s)/2}$) of RNA expression profiles across 50 tissues in

the Human Protein Atlas "RNA consensus tissue gene" data (from proteinatlas.org; version 23.0) (Uhlén et al, 2015). For subcellular localization, we first determined the pairwise distances of the subcellular location terms based on their overlap of annotated SLCs (Jaccard distance) and then integrated the distances of potentially multiple localizations per SLC–SLC pair to a single distance using the "best-match average" (BMA) method (Gu, 2024). For structure, we employed distances from a recent structural analysis of the solute carrier superfamily (Ferrada and Superti-Furga, 2022) which are based on the overlap of pairwise alignments of experimental or modeled protein structures. For substrate annotation, we calculated the distance between molecular substrates as the Tanimoto coefficient of OpenBabel FP4 fingerprints using the "ChemmineR" library (version 3.54.0) (Cao et al, 2008) on SDF format representations of the molecular compounds downloaded from ChEBI (version 22.9). The distance between elemental substrates was defined as Euclidean distance of the scaled "Oliynyk" representation in the "ElementEmbeddings" library (Onwuli et al, 2023). Elemental substrate distances were further scaled to a maximum of 1 and then combined with molecular substrate distances to an overall substrate distance matrix, using the maximum distance (i.e., $d = 1$) between molecular and elemental substrates. Again, the distances of potentially multiple substrates per SLC–SLC pair were integrated to a single distance using the BMA method. For disease associations, the semantic distance of two disease terms was determined from the graph representation in the Mondo ontology using the "Leacock" distance (Leacock and Chodorow, 1998) implemented in the "simona" library (version 1.0.10) (Gu, 2024). The distances of potentially multiple disease associations per SLC–SLC pair were integrated into a single distance using the BMA method. The eight modality-specific distances for SLC–SLC pairs varied considerably in range and distribution. Moreover, due to the chosen metrics, the distances for four modalities were continuously distributed, while distances for the other four modalities were truncated to the interval [0,1] (interaction proteomics, subcellular localization, substrate annotation, disease association).

To avoid any distribution-based bias in downstream integration, we transformed the distance distributions to dissimilarity distributions via an Ordered Quantile normalization using the R package "bestNormalize" (version 1.9.1) (Peterson, 2021). Values at the lowest and highest ends of the four truncated distance distributions were excluded from this transformation process. The resulting standard normally distributed dissimilarities were further transformed into a truncated normal distribution using the library "truncnorm" (version 1.0-9) (Mersmann et al, 2025) with $\mu = 0.5$ and $\sigma = 0.194$, chosen so that a corresponding normal distribution would cover 99% of the data in the interval between 0 and 1. Distances at the lower or upper limit from the four modalities with truncated distance distributions were now reintroduced at 0 or 1, respectively, resulting in modality-specific, but more comparable SLC–SLC dissimilarities (Fig. EV5B). Resulting dissimilarities were subtracted from 1 to result in modality-specific, but more comparable SLC–SLC similarities. To assess redundancy between the different dissimilarity measures, correlation using Pearson's method on pairwise-complete observations was calculated. The information content of each dissimilarity measure was estimated by calculating its alpha entropy using the library "bigdatadist" (version 1.1) (Martos et al, 2018). The dissimilarities were subtracted from 1 to result in similarities which were then combined into a single similarity

per SLC pair by a weighted average, putting the highest weight of 3 on structural similarity and the three main experimental data sets in this study (transcriptomics, metabolomics and interaction proteomics), medium weight of 2 on the substrate annotation, and the lowest weight of 1 on tissue expression, subcellular localization and disease association (Table 1; Fig. EV5E).

## Constructing the SLC landscape and graph-based clustering

The integrated, pairwise SLC similarities were subtracted from 1 to result in a dissimilarity matrix, which in turn was used as the input distance for the construction of a high-dimensional graph representation via the "umap-learn" library (version 0.5.3) (McInnes et al, 2018), with a local neighborhood size of 15. The graph was then embedded into a two-dimensional landscape using 500 training epochs and a minimum distance between embedded points of 0.1.

We derived SLC–SLC distances from the high-dimensional graph by defining the distance for each of its edges as its logarithmized adjacency value subtracted from 1, and adding up those edge distances along the shortest path connecting any pair of SLCs. Compared to the weighted-average pairwise SLC–SLC similarity, which was used as the input to the landscape construction, this graph-based distance incorporates local communities as well as global structures in the landscape.

To identify communities in the SLC landscape, clustering was performed on the high-dimensional graph using the Leiden method, with modularity partition, a resolution of 1.0 and 100 iterations, employing the "leiden" (version 0.4.3.1) and "leidenalg" (version 0.10.1) libraries (Traag et al, 2019).

For the cross-validation analysis, the landscape was reconstructed eight times, each time excluding a different modality.

## Analysis of SLC properties on the landscape

Distances on the landscape (graph-based distance) and original distances from the eight modalities were tested for coherence with the SLCs' single-label properties (family, fold, and substrate class) using the analysis of similarities (ANOSIM) method (Clarke, 1993) as implemented in the "vegan" library (version 2.6-4). The significance of coherence was assessed by 10,000 permutation rounds.

Drivers of clustering were identified by calculating the adjusted mutual information scores between SLC landscape clusters and their SLCs' single-label properties (family, fold, substrate class, number of modalities) using the "sklearn" library (version 1.3.2) in Python (version 3.11.6) (Fig. EV5H).

To find enriched single-label (family, fold, and substrate class) and multi-label (subcellular location) SLC properties in the clusters, we conducted one-sided Fisher's exact tests for each SLC property label in each cluster. Property labels featured by fewer than two SLCs were omitted. Resulting $p$ values were corrected for multiple testing following the Benjamini–Hochberg procedure (Benjamini and Hochberg, 1995) (Fig. EV5I,J).

## Data availability

The data sets produced in this study are available in the following databases: Immunofluorescence images: BioImage Archive S-BIAD1630

(https://doi.org/10.6019/S-BIAD1630) and RESOLUTE web portal (https://re-solute.eu/resources/datasets). The data sets and resources referenced in this study are available through the RESOLUTE web portal (https://re-solute.eu), the knowledgebase (https://re-solute.eu/knowledgebase), and the dashboards (https://re-solute.eu/resources).

The source data of this paper are collected in the following database record: biostudies:S-SCDT-10_1038-S44320-025-00108-2.

## Peer review information

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

## Acknowledgements

We would like to acknowledge Enrico Girardi for his early preparatory work, Gregor Redinger for support on web portal development, as well as the previous and current Superti-Furga lab members who screened the literature for reported SLC substrates, namely Abigail Jarret, Andras Böszörmenyi, Andrea Casiraghi, Ann-Katrin Hopp, Barbara Steurer, Benedikt Neumayer, Patrick Essletzbichler, Selene Glueck, and Vojtech Dvorak. This study received funding from the RESOLUTE and REsolution consortia. RESOLUTE has received funding from the Innovative Medicines Initiative 2 Joint Undertaking under grant agreement No. 777372. This Joint Undertaking receives support from the European Union's Horizon 2020 research and innovation programme and EFPIA. REsolution has received funding from the Innovative Medicines Initiative 2 Joint Undertaking under grant agreement No. 101034439. This Joint Undertaking receives support from the European Union's Horizon 2020 research and innovation programme and EFPIA. This article reflects only the authors' views and neither IMI nor the European Union and EFPIA are responsible for any use that may be made of the information contained therein. The last year of work, including validation of data and writing of the manuscript was supported mainly by the Austrian Academy of Sciences. BH was supported by funds from the Austrian Science Fund FWF (https://doi.org/10.55776/KLI1056 to K Boztug). GS-F was supported by the Austrian Academy of Sciences.

## Author contributions

**Ulrich Goldmann**: Conceptualization; Data curation; Software; Formal analysis; Supervision; Funding acquisition; Validation; Investigation; Visualization; Methodology; Writing—original draft; Project administration; Writing—review and editing. **Tabea Wiedmer**: Conceptualization; Data curation; Formal analysis; Supervision; Validation; Investigation; Visualization; Methodology; Writing—original draft; Project administration; Writing—review and editing. **Andrea Garofoli**: Data curation; Formal analysis; Investigation; Visualization; Methodology; Writing—original draft; Writing—review and editing. **Vitaly Sedlyarov**: Software; Methodology. **Manuel Bichler**: Software; Methodology. **Ben Haladik**: Formal analysis; Investigation; Writing—review and editing. **Gernot Wolf**: Validation. **Eirini Christodoulaki**: Validation. **Alvaro Ingles-Prieto**: Validation; Visualization. **Evandro Ferrada**: Data curation; Formal analysis;

Investigation; Methodology; Writing—original draft; Writing—review and editing. **Fabian Frommelt**: Validation. **Shao Thing Teoh**: Validation; Writing—review and editing. **Philipp Leippe**: Validation. **Gabriel Onea**: Investigation; Writing—original draft; Writing—review and editing. **Martin Pfeifer**: Data curation; Formal analysis; Investigation; Methodology; Writing—review and editing. **Mariah Kohlbrenner**: Investigation; Methodology. **Lena Chang**: Investigation; Methodology. **Paul Selzer**: Supervision. **Jürgen Reinhardt**: Resources; Supervision; Funding acquisition. **Daniela Digles**: Data curation. **Gerhard F Ecker**: Supervision; Funding acquisition. **Tanja Osthushenrich**: Data curation; Formal analysis; Investigation; Methodology; Writing—original draft; Writing—review and editing. **Aidan MacNamara**: Supervision; Funding acquisition; Investigation; Methodology. **Anders Malarstig**: Supervision; Funding acquisition; Investigation; Methodology. **David Hepworth**: Conceptualization; Supervision; Funding acquisition. **Giulio Superti-Furga**: Conceptualization; Resources; Supervision; Funding acquisition; Methodology; Writing—original draft; Project administration; Writing—review and editing.

Source data underlying figure panels in this paper may have individual authorship assigned. Where available, figure panel/source data authorship is listed in the following database record: biostudies:S-SCDT-10_1038-S44320-025-00108-2.

## Disclosure and competing interests statement

GS-F is a co-founder and owns shares of Solgate GmbH, an SLC-focused company. DH and AM are employees and stockholders of Pfizer.

# Expanded View Figures

**Figure EV1.  SLC expression and substrate annotation.**

(**A**) Number of SLC genes expressed in relation to the total number of protein-coding genes expressed in 1206 human cell lines (data from (Uhlén et al, 2015)). The dotted line indicates the overall average of 256 expressed SLCs per cell line (56.2% of the 455 SLCs with expression data). The dashed line indicates the average of 2.2% of SLC genes among all expressed protein-coding genes in a cell line. Cell lines of selected origin are indicated by color, and the cell lines employed in the RESOLUTE consortium are labeled. (**B**) Number of SLC genes expressed in relation to the total number of protein-coding genes expressed in 50 human tissues (data from (Uhlén et al, 2015)). The dotted line indicates the overall average of 300 expressed SLCs per tissue (65.9% of the 455 SLCs with expression data). The dashed line indicates the average of 2.2% of SLC genes among all expressed protein-coding genes in a tissue. Tissue groups, as defined in the original data, are indicated by color and selected tissues deviating from the overall distribution are labeled. (**C**) Total number of SLC genes expressed in 36 tissue groups. The average is indicated by the dotted line. Classification of tissue specificity of gene expression is indicated by color (data from (Uhlén et al, 2015)). (**D**) Comparison of previous (Meixner et al, 2020) and updated SLC substrate classifications with regards to the terminology and membership of SLCs. The width of the connecting lines corresponds to the number of co-classified SLCs. (**E**) The proportion of SLC families for each substrate class, i.e., number of respective family members compared to the total number of SLCs in each substrate class. The total numbers are given in parentheses.

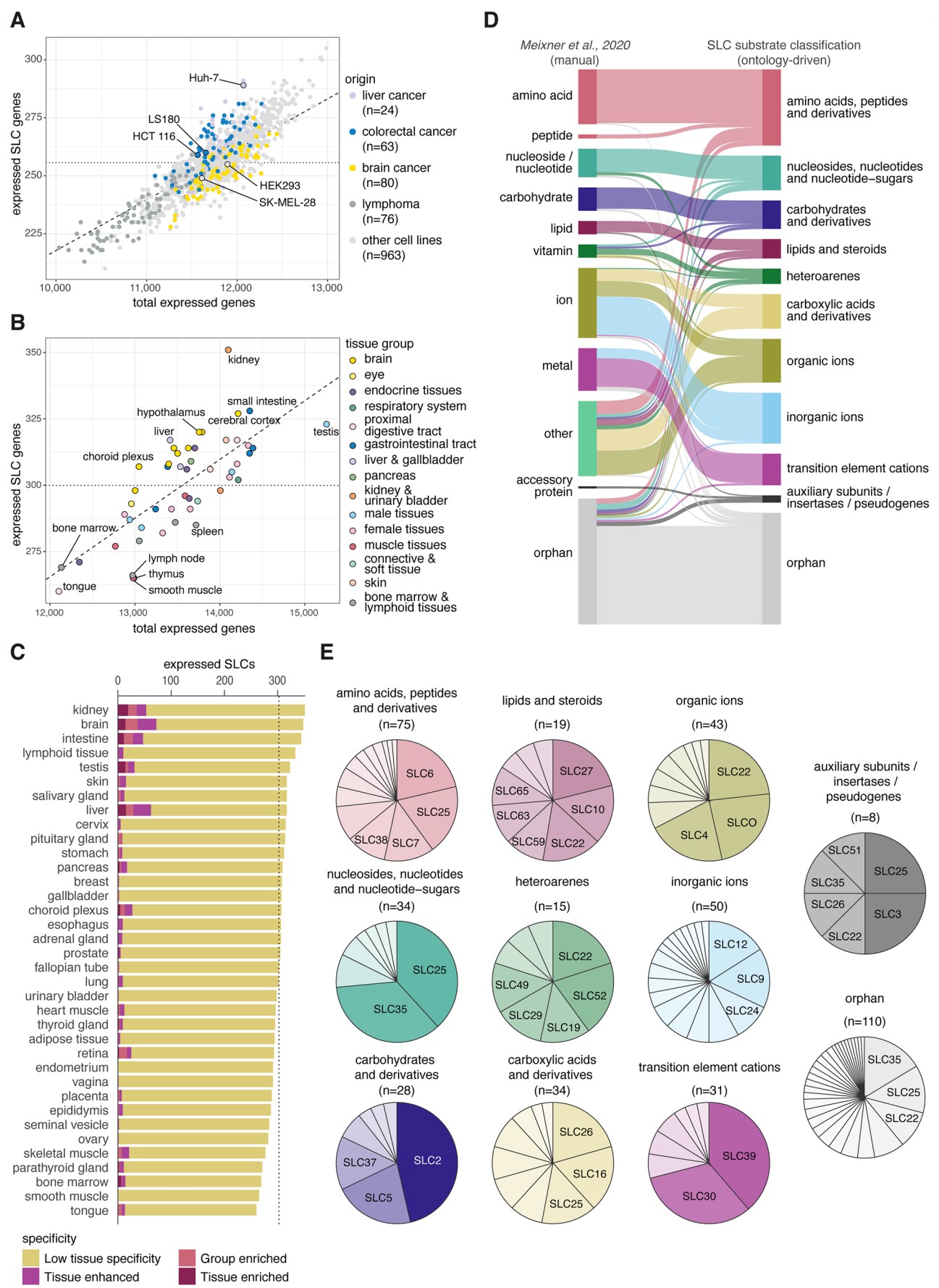

**A**

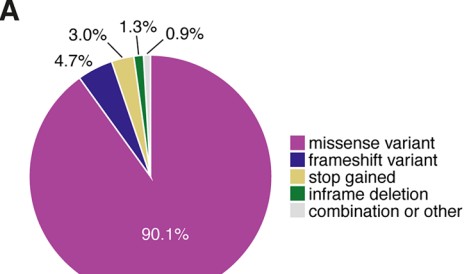

3.0%  1.3%  0.9%
4.7%

90.1%

- missense variant
- frameshift variant
- stop gained
- inframe deletion
- combination or other

**B**

| ontology of mapped trait | variant-level associations (pathogenic) | gene-level associations (pathogenic) | distinct gene-term associations (pathogenic) |
|---|---|---|---|
| **MONDO** | 3,475 (3,244) | 218 (216) | 946 (780) |
| **EFO** | 4,035 (0) | 261 (0) | 2,911 (0) |
| **HP** | 303 (0) | 5 (0) | 245 (0) |
| **GO** | 3 (0) | 1 (0) | 3 (0) |
| **total** | 7,816 (3,244) | 485 (216) | 4,105 (780) |

**C**

68 SLCs

127 pathogenic terms

**Figure EV2. Collection of SLC genetic variants and disease associations.**

(A) Proportions of the main types of genetic alterations of variants with a protein-altering consequence on the canonical transcript of an SLC gene. (B) Overview of the number of variants with trait mapping to ontologies Mondo, EFO, HP and GO. Numbers in parentheses are pathogenic variants. (C) Heatmap of 68 SLCs without pathogenic associations in ClinVar and Orphanet (x axis) and novel 127 putative pathogenic terms (y axis). Examples discussed are marked in bold and light blue.

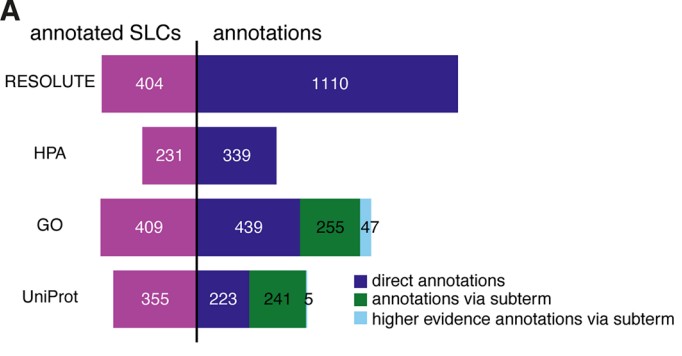

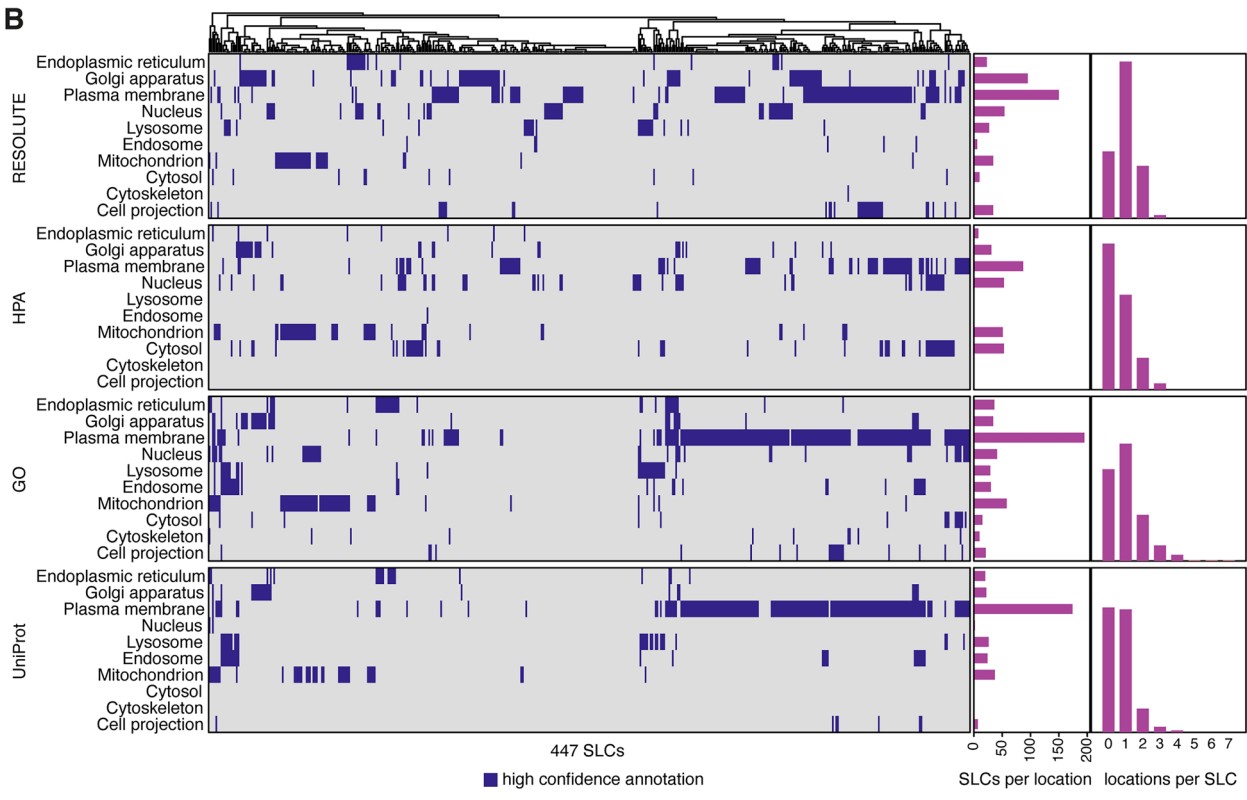

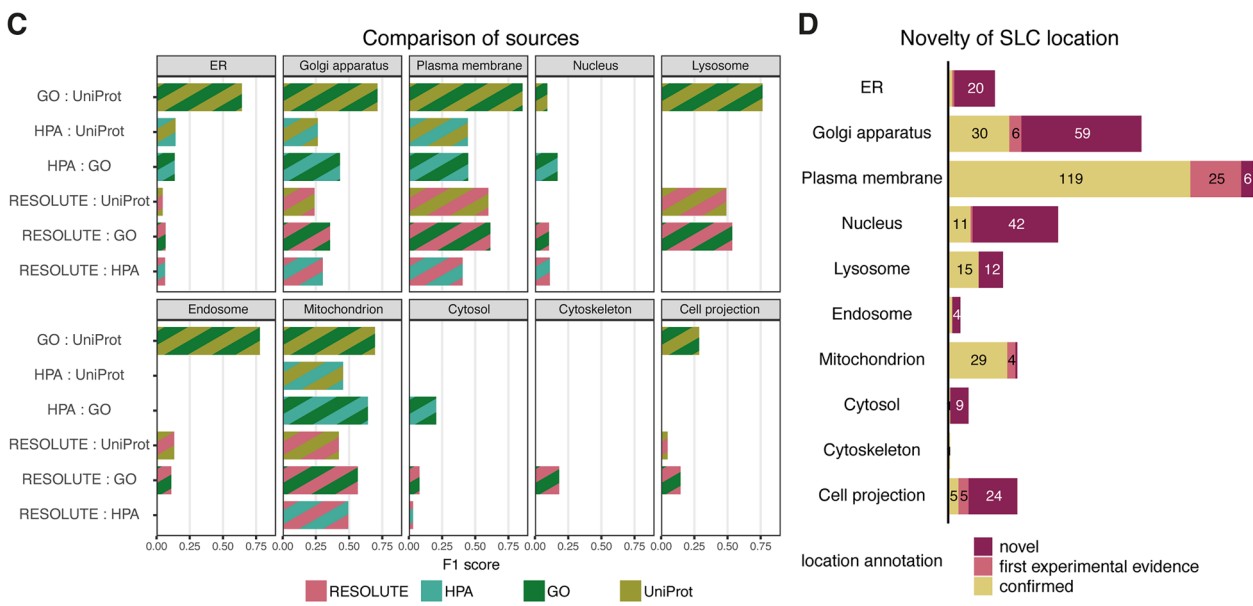

◀ **Figure EV3. Comparison of subcellular location annotations for the SLC superfamily.**

(A) Comparison of the number of location annotations by resource (as in Fig. 3C), highlighting the effect of the described ontology matching process (Methods). For GO and UniProt, a considerable number of annotations were only captured via subterms of the 10 selected locations. A smaller number of location annotations did match a selected location term but ended up with an increased evidence level via an additionally mapped subterm of the respective location. (B) Overview and comparison of all high-confidence location annotations across RESOLUTE, HPA, GO and UniProt. The histograms show the prevalence of annotations for each of the selected 10 locations per data source, as well as the distribution of the number of location annotations per SLC for each data source. (C) Scoring of the agreement in annotations per subcellular location, for each pair of annotation sources. The F1-score combines precision and recall between the two annotation sources, with a higher score corresponding to higher consistency. (D) Novelty of high-confidence experimental localization annotations by the RESOLUTE data set, for each of the 10 selected subcellular locations.

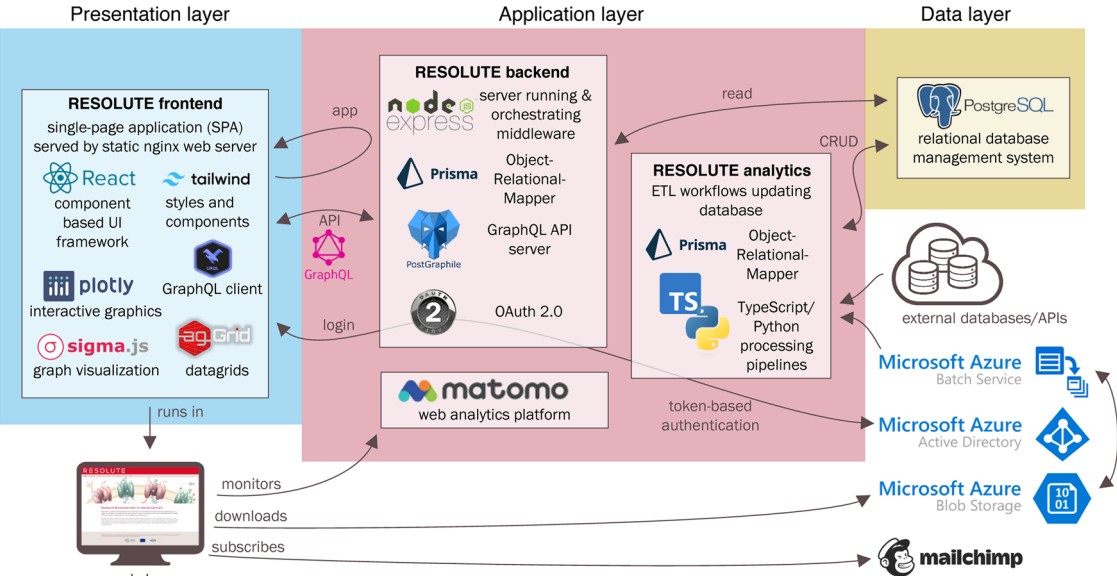

**Figure EV4. The RESOLUTE web portal software architecture.**

The web portal features a relational database handling the data layer, a backend and processing workflows handling the application layer, and a frontend in the form of a 'single-page' web application handling the presentation layer. Please refer to the Methods section for more details. UI user interface, API application programming interface, CRUD create, read, update, delete.

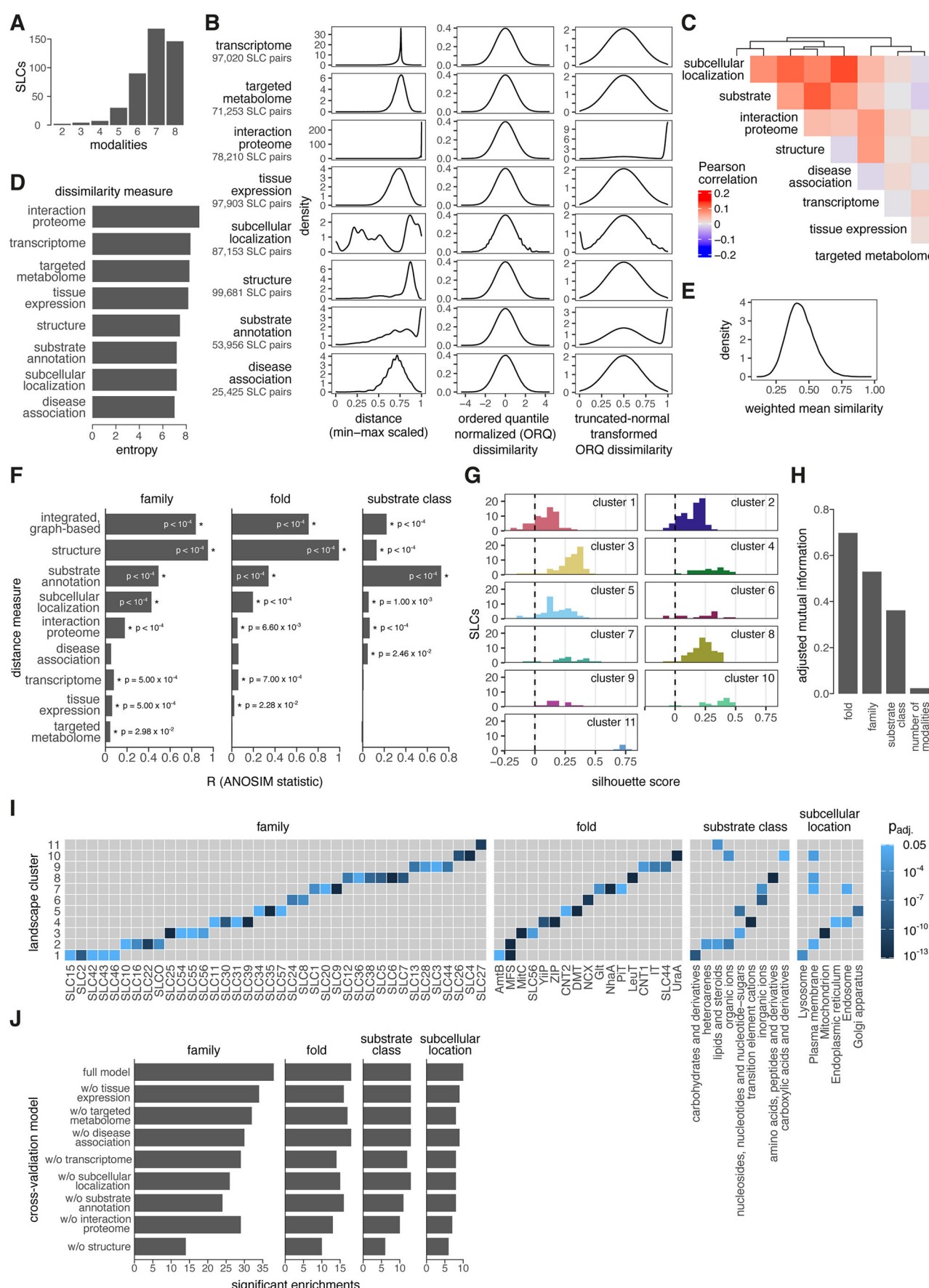

◀ **Figure EV5.   Data sets used in constructing the SLC landscape and analysis of its clustering.**

(**A**) Distribution of the number of modalities (data sets) available per SLC. (**B**) Distribution of SLC–SLC pair distances per modality. The original distributions of modality-specific distances are shown in the left column, scaled to the interval of [0,1] for visualization purposes. The central column shows the same distributions after transformation to a standard normal distribution using ordered quantile normalization. For distances with an upper or lower limit, values at these limits were excluded from this transformation. The right column shows the normalized distributions transformed to a truncated normal distribution, with the previously excluded values reintroduced at the corresponding boundary, resulting in dissimilarities comparable across modalities. (**C**) Correlations between the eight dissimilarities of all possible SLC–SLC pairs ($n = 99,681$), using Pearson's method on pairwise-complete observations. (**D**) Information content estimation for each of the eight dissimilarities, computed by the alpha entropy for continuous values. (**E**) Distribution of the overall SLC–SLC pair similarities, which resulted from subtracting the weighted average of up to eight dissimilarities for each pair from 1. (**F**) ANOSIM analysis for coherence of different distances (original distances and integrated landscape distance) with discrete SLC properties (family, fold, substrate class). Significant coherences ($P < 0.05$) are marked with an asterisk and the corresponding $P$ value from a permutation-based test. Due to the test involving 10,000 iterations, $P$ values below $10^{-4}$ cannot be accurately determined. (**G**) Distribution of silhouette scores of the members of the 11 different clusters derived from the SLC landscape. (**H**) Mutual information analysis of landscape clusters with discrete SLC properties. Fold, family, and substrate class all share considerable mutual information with the 11 clusters of the SLC landscape. The number of modalities does not correspond to cluster membership. (**I**) Results of the pairwise enrichment analysis of each SLC property level in each cluster, using Fisher's exact test. Only SLC property levels with significant enrichments are shown, colored by their multiple-testing corrected $P$ values. (**J**) Summary of the enrichment analysis in panel I ('full model'), and additionally computed for SLC landscape clusterings generated by a leave-one-out cross-validation analysis. Models are ordered by the total number of enrichments per round of cross-validation. The full model not only shows the highest number of enrichments overall, but also in each SLC property analyzed.

