## [Peer Review File · Molecular Systems Biology]

Data- and knowledge-derived functional landscape of human solute carriers

Ulrich Goldmann, Tabea Wiedmer, Andrea Garofoli, Vitaly Sedlyarov, Manuel Bichler, Ben Haladik, Gernot Wolf, Eirini Christodoulaki, Alvaro Ingles-Prieto, Evandro Ferrada, Fabian Frommelt, Shao Teoh, Philipp Leippe, Gabriel Onea, Martin Pfeifer, Mariah Kohlbrenner, Lena Chang, Paul Selzer, Juergen Reinhardt, Daniela Digles, Gerhard F Ecker, Tanja Osthusenrich, Aidan MacNamara, Anders Malarstig, David Hepworth, and Giulio Superti-Furga

Corresponding author(s): Giulio Superti-Furga (gsupert@cemm.oeaw.ac.at)

Review Timeline:

Submission Date:	17th Oct 24
Editorial Decision:	29th Nov 24
Revision Received:	30th Jan 25
Editorial Decision:	11th Mar 25
Revision Received:	28th Mar 25
Accepted:	11th Apr 25

Editor: Poonam Bheda

Transaction Report:

29th Nov 2024

Manuscript Number: MSB-2024-12692

Title: Data- and knowledge-derived functional landscape of human solute carriers

Dear Dr. Superti-Furga,

Thank you for the submission of your manuscript to Molecular Systems Biology. We have now received feedback from the three reviewers who agreed to evaluate your manuscript. As you will see from the reports below, the referees acknowledge the interest of the study and are overall supporting publication of your work pending appropriate revisions.

Without repeating all the comments listed below, one of the more fundamental issues raised by Reviewer 3 is that the revisions should focus on justification, validation, and comparison with other clustering methods. All other issues raised would need to be satisfactorily addressed.

On a more editorial level and in line with comments from Reviewer 2, we would ask you to work to improve the writing and presentation of the manuscript. Please let me know in case you would like to discuss thir or any of the comments in further detail, I would be happy to schedule a call.

We require:

1) A .docx formatted version of the manuscript text (including legends for main figures, EV figures and tables). Please make sure that the changes are highlighted to be clearly visible. Alternatively you may choose to submit your manuscript as a LaTeX file.

4) A .docx formatted letter INCLUDING the reviewers' reports and your detailed point-by-point responses to their comments. As part of the EMBO Press transparent editorial process, the point-by-point response is part of the Peer Review File (PRF), which will be published alongside your paper.

5) A complete author checklist, which you can download from our author guidelines (<https://www.embopress.org/page/journal/17574684/authorguide#submissionofrevisions>). Please insert information in the checklist that is also reflected in the manuscript. The completed author checklist will also be part of the PRF.

6) Please note that all corresponding authors are required to supply an ORCID ID for their name upon submission of a revised manuscript.

7) It is mandatory to include a 'Data Availability' section after the Materials and Methods. Before submitting your revision, primary datasets produced in this study need to be deposited in an appropriate public database, and the accession numbers and database listed under 'Data Availability'. Please remember to provide a reviewer password if the datasets are not yet public (see <https://www.embopress.org/page/journal/17574684/authorguide#dataavailability>).

In case you have no data that requires deposition in a public database, please state so in this section. Note that the Data Availability Section is restricted to new primary data that are part of this study. This study includes no data deposited in external repositories.

8) All Materials and Methods need to be described in the main text using our 'Structured Methods' format, which is required for all research articles. According to this format, the Methods section includes a Reagents and Tools Table (listing key reagents, experimental models, software and relevant equipment and including their sources and relevant identifiers) followed by a Methods and Protocols section describing the methods using a step-by-step protocol format. The aim is to facilitate adoption of the methodologies across labs. Please upload the Reagents and Tools table as a separate document when submitting your revised manuscript. More information on how to adhere to this format as well as a downloadable template (.docx) for the Reagents and Tools Table can be found in our author guidelines:

<https://www.embopress.org/page/journal/17444292/authorguide#structuredmethods>

An example of a Method paper with Structured Methods can be found here:
<https://www.embopress.org/doi/10.15252/msb.20178071>.

9) For data quantification: please specify the name of the statistical test used to generate error bars and P values, the number (n) of independent experiments (specify technical or biological replicates) underlying each data point and the test used to calculate p-values in each figure legend. The figure legends should contain a basic description of n, P and the test applied. Graphs must include a description of the bars and the error bars (s.d., s.e.m.). Please provide exact p values.

10) Our journal encourages inclusion of *data citations in the reference list* to directly cite datasets that were re-used and obtained from public databases. Data citations in the article text are distinct from normal bibliographical citations and should directly link to the database records from which the data can be accessed. In the main text, data citations are formatted as follows: "Data ref: Smith et al, 2001" or "Data ref: NCBI Sequence Read Archive PRJNA342805, 2017". In the Reference list, data citations must be labeled with "[DATASET]". A data reference must provide the database name, accession number/identifiers and a resolvable link to the landing page from which the data can be accessed at the end of the reference. Further instructions are available at .

11) We replaced Supplementary Information with Expanded View (EV) Figures and Tables that are collapsible/expandable online. A maximum of 5 EV Figures can be typeset. EV Figures should be cited as 'Figure EV1, Figure EV2' etc... in the text and their respective legends should be included in the main text after the legends of regular figures.

<https://www.embopress.org/page/journal/17574684/authorguide#expandedview>

13) Author contributions: CRedit has replaced the traditional author contributions section because it offers a systematic machine readable author contributions format that allows for more effective research assessment. Please remove the Authors Contributions from the manuscript and use the free text boxes beneath each contributing author's name in our system to add specific details on the author's contribution. More information is available in our guide to authors.

Please also suggest a striking image or visual abstract to illustrate your article as a PNG file 550 px wide x 300-600 px high. Share synopsis text and image, as well as eTOC:

Please note that these would be the final versions and changes during proofing are usually not allowed

16) As part of the EMBO Publications transparent editorial process initiative (see our policy here:

https://www.embopress.org/transparent-process#Review_Process), Molecular Systems Biology will publish online a Peer Review File (PRF) to accompany accepted manuscripts.

In the event of acceptance, this file will be published in conjunction with your paper and will include the anonymous referee reports, your point-by-point response and all pertinent correspondence relating to the manuscript. Let us know whether you agree with the publication of the PRF and as here, if you want to remove or not any figures from it prior to publication.

Please note that the Authors checklist will be published at the end of the PRF.

Molecular Systems Biology has a "scooping protection" policy, whereby similar findings that are published by others during review or revision are not a criterion for rejection. Should you decide to submit a revised version, I do ask that you get in touch after three months if you have not completed it, to update us on the status.

I look forward to receiving your revised manuscript.

Yours sincerely,

Poonam Bheda, PhD
Scientific Editor
Molecular Systems Biology

Reviewer #1:

Goldmann et al present in this manuscript an integration of knowledge on human solute carrier (SLC) proteins, including their structural fold, subcellular localization and disease associations. A website portal is provided, which consolidates data curated from the literature as well as results generated by the RESOLUTE and REsolution consortium. Personally I find this effort both important and useful, and the manuscript is well-written with good amounts of valuable insights. I strongly recommend its publication, though it might be further improved by addressing the following two general comments.

1) The authors seem to overlook other contemporary efforts in the systematic study of SLCs, which sometimes makes it unclear whether there is consensus on certain aspects. For example, the manuscript classifies SLCs by structural fold (Figure 1A), citing the iScience 2022 paper. How does this classification compare to that in the 2022 Structure paper (Structure, 30, 1321)? The authors should comment on this in the manuscript.

2) The similarity profiles can be useful but the eight dimensions seem imbalance in terms of the information contained, considering those contained in transcriptome and in subcellular localization. How is this imbalance handled? Additionally, similarity could also be assessed using latent features from protein language models (PLMs). How would this approach compare with the SLC-SLC similarities proposed in this study? Please comment.

Reviewer #2:

SLCs are a major group of transport proteins with important functions in a range of cellular processes. Many SLCs have unannotated substrates and/or functions. The Resolute consortium has been attempting to collect a range of functional and biochemical data on SLCs thus hoping to better characterise them. The current manuscript represents a component of this analysis in particular the integration of data collected in other studies by the consortium (apparently described in related manuscripts) and previous knowledge culminating in production of a searchable database.

Overall the work represents a significant effort to produce robust and comprehensive analyses. The figures describing this are mostly clear and convey useful and interesting insights. There are some well-chosen examples in the text that illustrate what kind of information can be gleaned. The major output of the MS, of course, is the online database which appears to work well and is aesthetically pleasing. The MS is thus a key resource for the community and should certainly be published.

Whilst the data and content of the MS is superb, the writing in the main body of the text could be improved. There are a number of grammatical errors, but more importantly there are many sections where the writing is confusing and needs clarification. Additionally, some sections are disorganised, with information being repeated in different, unconnected paragraphs. The discussion, whilst containing some excellent sections, also contains some rather grandiose sections which don't add much, giving the overall impression that the discussion was written by a committee rather than providing a coherent overview of the work's implications. I think it would improve uptake of the resource if the text was as pleasing as the figures (and the excellent website).

It's not feasible for me to go through and edit the language in all places, but I would encourage the authors to really check thoroughly through the whole manuscript. It's particularly important to look at the abstract, which could do with some rewriting, and the introduction which has a few bits that are difficult to understand.

In terms of the analyses I have a few small points where I think a modest amount of reanalysis could be helpful- see the Analysis Comments section below.

Analysis comments

1) The authors have heroically curated SLC data from the literature, which is excellent. I note that another recent work has done something similar (doi: <https://www.biorxiv.org/content/10.1101/2024.06.24.600425v1>) to get known SLC/substrate pairs. Could the authors investigate the extent to which their annotations agree with those in this manuscript? This would be useful in testing the robustness of the literature curation in the two studies.

2) In the disease association figure, panel C is not very intuitive. Could the authors replace this with a simple histogram showing

the number of disease associations per SLC on the x axis and the frequency of this on the y axis? I think this would be easier for the reader to use to understand how common multiple or single disease annotations are across SLCs. It would be good to include as a bar the number of SLCs that have no annotated or predicted disease association (i.e. the x axis should begin at 0).

3) The website is excellent and the authors have provided flat files with key data in the supplement. However, it would be very important for the authors to also produce a flat file that contains the output of the website search for each SLC as a single file (even though it would be large) which could be a supplemental table or downloadable from the website. This would make it easier for bioinformaticians to perform global analyses on the entire dataset rather than having to curate the individual flat files from the supplement or search individually using the website for a large number of SLCs individually. This would increase the utility of the resource considerably.

Reviewer #3:

The manuscript "Data- and knowledge-derived functional landscape of human solute carriers" integrates multimodal data of SLCs (structure, expression, substrates, disease associations, experimental annotation of subcellular location, transcriptional and metabolic data), and supplements and filters SLC-related data from public databases. Eight features were selected for multimodal clustering to identify potential functional regions, and a series of descriptive analyses were conducted. However, the authors need to address the following issues:

- 1.Regarding the semi-automated substrate assignment method for SLCs: Is this method appropriate? Is there a specific order for categorizing the substrates? What is the rationale behind the chosen order?
- 2.Alternative methods for clustering the similarity matrix: Are there other clustering methods that could be used? Why was the UMAP algorithm chosen for dimensionality reduction and clustering in this study?
- 3.Weight determination in multimodal data fusion: How are the weights assigned to each modality during the fusion of multimodal data? What is the theoretical basis for this weighting scheme?
- 4.Ensuring the validity of multimodal data: How can the validity of the integrated multimodal data be ensured to prevent noise? Can an ablation study be performed to assess the contribution of each modality and validate the effectiveness of the data integration?
- 5.Validating the effectiveness of the multimodal clustering method: Can an experiment be designed to validate the effectiveness of the multimodal clustering method, rather than relying solely on descriptive analysis?
- 6.Noise in multimodal clustering: Given the introduction of multiple modalities, is there a risk of noise being generated in the clustering results due to inconsistencies or redundancies between the data types? How can the accuracy of the results be ensured?
- 7.Comparison with existing methods: Is it necessary to compare this method with existing clustering approaches to assess its relative advantages and limitations?

Response to reviewers of “Data- and knowledge-derived functional landscape of human solute carriers”

We think that the addition of three new analyses and the improved writing has significantly improved the manuscript. We thank the reviewers for their critical but constructive comments and suggestions.

Summary of key points included in this revision

- Added an analysis of the information content to explain and justify the weighted integration of the eight similarity measures used.
- Added a statistical analysis of the individual distances and the integrated landscape distance to demonstrate the validity of the UMAP-based integration method.
- Performed a leave-one-out cross-validation analysis to confirm effectiveness of the integration and clustering method.
- Improved the writing and clarified confusing or repetitive sections throughout the manuscript.

We added a new Figure 1A to show the complementarity of this and the three accompanying large-scale studies on SLCs.

Below, the reviewers' comments are represented in black, followed by our responses in green. To simplify revision, we added page and line numbers referring to the **clean** document or to the '**track changes**' document in the *simple markup* view.

Point-by-point response – Reviewer #1:

Goldmann et al present in this manuscript an integration of knowledge on human solute carrier (SLC) proteins, including their structural fold, subcellular localization and disease associations. A website portal is provided, which consolidates data curated from the literature as well as results generated by the RESOLUTE and REsolution consortium. Personally I find this effort both important and useful, and the manuscript is well-written with good amounts of valuable insights. I strongly recommend its publication, though it might be further improved by addressing the following two general comments.

1. The authors seem to overlook other contemporary efforts in the systematic study of SLCs, which sometimes makes it unclear whether there is consensus on certain aspects. For example, the manuscript classifies SLCs by structural fold (Figure 1A), citing the iScience 2022 paper. How does this classification compare to that in the 2022 Structure paper (Structure, 30, 1321)? The authors should comment on this in the manuscript.

We thank the reviewer for pointing us at the specific study of Xie *et al.* on SLC structural folds. The work presented there is indeed very similar to the one by Ferrada *et al.* and we added a reference to Xie *et al.* in the introduction on SLC structural folds accordingly.

For integration in our functional landscape, we rely on similarities between pairs of SLCs. And while Xie *et al.* apparently did calculate these similarities (as can be seen in the dendrogram in their Figure 4) they unfortunately did not share this information with their readers in form of supplemental material. A similarity table, the predicted SLC protein structures and even the fold classifications are all not available, and therefore we decided to keep using the data published by Ferrada *et al.* for our integration and classification purposes.

Changes: Adapted introduction on SLC folds and added reference to Xie et al. [page 2, lines 66-68].

2. The similarity profiles can be useful but the eight dimensions seem imbalance in terms of the information contained, considering those contained in transcriptome and in subcellular localization. How is this imbalance handled? Additionally, similarity could also be assessed using latent features from protein language models (PLMs). How would this approach compare with the SLC-SLC similarities proposed in this study? Please comment.

The eight modalities selected for integration to a functional SLC landscape are indeed expected to be different in information content. We subjectively assigned different weights for the integration, handling the imbalance in information but foremost reflecting the composition of function that we want our landscape to represent. Our priority on large scale data-creation is also based on extensive discussions on the research strategy of RESOLUTE that itself followed several international meetings on research priorities on SLC transporters.

Based on the reviewer's question, we performed an additional analysis to more objectively assess the information content of each modality based on *alpha entropy* (Martos *et al.*, 2018). Indeed, the resulting ranking of information content largely corresponded to our weighting scheme. The experimental data sets of interaction proteomics (weight: 3), transcriptomics (weight: 3) and metabolomics (weight: 3) showed the highest information content, followed by tissue expression (weight: 1), structure (weight: 3) and the annotation-based data sets for substrate (weight: 2), localization (weight: 1) and disease association (weight: 1). However, we think that structure and substrate annotation represent essential properties of an SLC transporter and therefore decided to keep the weights for these two modalities unchanged.

In the revised manuscript, we added more detail on the weighting and integration process to the result section, and a figure panel with the results of the entropy calculations. The addition of this analysis improved the quality and clarity of our integration approach. We thank the reviewer for provoking this analysis.

Changes: added new Figure panel EV5D [page 32], adapted results section [page 10, lines 414-417] and added the entropy analysis to the methods section [page 38, lines 959-961].

We also had a look at protein language models, by employing the Evolutionary Scale Modeling (ESM-2; <https://github.com/facebookresearch/esm>) approach to generate a 2560-dimensional embedding for each SLC sequence. Pairwise Euclidean distances based on these embeddings showed correlation to the distances we derived from substrate annotation and structure, and to a lesser extent also to the distances we derived from subcellular localization annotation and interaction proteomics data (see Reviewer Figure 1, below). Thus, the ESM embeddings clearly capture aspects of protein function, which is not surprising as ESM was developed not only for predicting protein structures but also for functional protein design (<https://doi.org/10.1073/pnas.2016239118>).

Point-by-point response for the manuscript: Data- and knowledge-derived functional landscape of human solute carriers.

Reviewer Figure 1. Correlation of SLC-SLC pairwise Euclidean distance of ESM-2 embeddings to other distance measures.

However, we decided not to include this PLM as a ninth modality since we want the landscape to stay interpretable, as we demonstrated in the example of Figure 5G. We think that finding another SLC close to one's SLC of interest based on a high similarity in their PLM embeddings is not as informative as the 8 modalities chosen.

Point-by-point response – Reviewer #2:

SLCs are a major group of transport proteins with important functions in a range of cellular processes. Many SLCs have unannotated substrates and/or functions. The Resolute consortium has been attempting to collect a range of functional and biochemical data on SLCs thus hoping to better characterise them. The current manuscript represents a component of this analysis in particular the integration of data collected in other studies by the consortium (apparently described in related manuscripts) and previous knowledge culminating in production of a searchable database.

Overall the work represents a significant effort to produce robust and comprehensive analyses. The figures describing this are mostly clear and convey useful and interesting insights. There are some well-chosen examples in the text that illustrate what kind of information can be gleaned. The major output of the MS, of course, is the online database which appears to work well and is aesthetically pleasing. The MS is thus a key resource for the community and should certainly be published.

Whilst the data and content of the MS is superb, the writing in the main body of the text could be improved. There are a number of grammatical errors, but more importantly there are many sections where the writing is confusing and needs clarification. Additionally, some sections are disorganised, with information being repeated in different, unconnected paragraphs. The discussion, whilst containing some excellent sections, also contains some rather grandiose sections which don't add much, giving the overall impression that the discussion was written by a committee rather than providing a coherent overview of the work's implications. I think it would improve uptake of the resource if the text was as pleasing as the figures (and the excellent website).

It's not feasible for me to go through and edit the language in all places, but I would encourage the authors to really check thoroughly through the whole manuscript. It's particularly important to look at the abstract, which could do with some rewriting, and the introduction which has a few bits that are difficult to understand.

We thank the reviewer for this encouraging assessment. We corrected grammatical errors throughout the manuscript and worked on clarification of confusing parts. We rewrote the abstract, removed repetitions from the result sections on substrate annotation and disease association, and streamlined the introduction and discussion.

In terms of the analyses I have a few small points where I think a modest amount of reanalysis could be helpful- see the Analysis Comments section below.

Analysis comments

1. The authors have heroically curated SLC data from the literature, which is excellent. I note that another recent work has done something similar (doi: <https://www.biorxiv.org/content/10.1101/2024.06.24.600425v1>) to get known SLC/substrate pairs. Could the authors investigate the extent to which their annotations agree with those in this manuscript? THIS would be useful in testing the robustness of the literature curation in the two studies.

The preprint by Zhang *et al.* (2024) used the initial substrate annotation work described in Meixner *et al.* (2020) to develop predictions for substrates of orphan SLCs. In Table S14, Zhang *et al.* provided additional literature-derived substrate annotations for 28 orphan SLCs.

In our study, the updated substrate annotation featured detailed, strict inclusion criteria (see Methods) and we already presented a comparison to the annotation in Meixner *et al.* (see Figure EV1D). While there is an overlap in substrates for 16 of the 28 additional annotated SLCs by Zhang *et al.*, we consider it wise to refrain from including a comparison in our manuscript, mainly due to a lack of detailed inclusion criteria in their non-peer reviewed work.

2. In the disease association figure, panel C is not very intuitive. Could the authors replace this with a simple histogram showing the number of disease associations per SLC on the x axis and the frequency of this on the y axis? I think this would be easier for the reader to use to understand how common multiple or single disease annotations are across SLCs. It would be good to include as a bar the number of SLCs that have no annotated or predicted disease association (i.e. the x axis should begin at 0).

We agree with the reviewer that the double-logarithmic plot in Fig.2C might not be as intuitive as we originally hoped. We followed the reviewer's suggestion and replaced the family-level disease association analysis with a hopefully more intuitive histogram of the disease associations per SLC.

Changes: Restructured Figure 2 and updated panel B (previous panel C) [page 20]. Adapted result section on disease association [page 6, lines 233-236].

3. The website is excellent and the authors have provided flat files with key data in the supplement. However, it would be very important for the authors to also produce a flat file that contains the output of the website search for each SLC as a single file (even though it would be large) which could be a supplemental table or downloadable from the website. THIS would make it easier for bioinformaticians to perform global analyses on the entire dataset rather than having to curate

the individual flat files from the supplement or search individually using the website for a large number of SLCs individually. This would increase the utility of the resource considerably.

We thank the reviewer for this remark and accordingly added downloadable tables of compiled annotation and experimental data to the “Resources” section (<https://re-solute.eu/resources/datasets>).

To further increase the web portal’s utility, especially for bioinformaticians and data scientists, we also keep improving API access to all the data sets (<https://re-solute.eu/graphiql>).

Point-by-point response – Reviewer #3:

The manuscript "Data- and knowledge-derived functional landscape of human solute carriers" integrates multimodal data of SLCs (structure, expression, substrates, disease associations, experimental annotation of subcellular location, transcriptional and metabolic data), and supplements and filters SLC-related data from public databases. Eight features were selected for multimodal clustering to identify potential functional regions, and a series of descriptive analyses were conducted. However, the authors need to address the following issues:

1. Regarding the semi-automated substrate assignment method for SLCs: Is this method appropriate? Is there a specific order for categorizing the substrates? What is the rationale behind the chosen order?

The reviewer refers to the inherently ambiguous task of assigning a single substrate-class to an SLC based on its annotated substrates and class labels derived from the chemical ontology tree. The order of substrate classes is following from more specific to more generic classes so that if a substrate of an SLC could be e.g. an amino acid but at the same time also an organic ion, the more specific class of amino acid was assigned. However, some substrate classes had to be manually curated, such as SLC16A12 of the “SLC16 monocarboxylate transporter family” for example, where one of its substrates is L-glutamine which classifies as amino acid, but also as carboxylic acid as most of its other substrates.

While the specific order used in this semi-automatic classification was already present in the Methods section, we expanded the respective paragraph to make this process more transparent.

Changes: Updated methods section on substrate class definitions [page 35, lines 818-822].

2. Alternative methods for clustering the similarity matrix: Are there other clustering methods that could be used? Why was the UMAP algorithm chosen for dimensionality reduction and clustering in this study?

The aim of dimensionality reduction was to infer a landscape that features global as well as local structures in the underlying similarity matrix. For this approach we consider UMAP state-of-the-art, and it provided satisfactory results. Therefore, we did not see the need for benchmarking different dimensionality reduction methods.

3. Weight determination in multimodal data fusion: How are the weights assigned to each modality during the fusion of multimodal data? What is the theoretical basis for this weighting scheme?

For integration, we subjectively assigned different weights to the modalities, handling the imbalance in information but foremost reflecting the composition of function that we want our landscape to represent. Our priority on large scale data-creation is also based on extensive discussions on the research strategy of RESOLUTE that itself followed several international meetings on research priorities on SLC transporters.

Based on the reviewer's question, we performed an additional analysis to more objectively assess the information content of each modality based on *alpha entropy* (Martos *et al.*, 2018). Indeed, the resulting ranking of information content largely corresponded to our weighting scheme. The experimental data sets of interaction proteomics (weight: 3), transcriptomics (weight: 3) and metabolomics (weight: 3) showed the highest information content, followed by tissue expression (weight: 1), structure (weight: 3) and the annotation-based data sets for substrate (weight: 2), localization (weight: 1) and disease association (weight: 1). However, we think that structure and substrate annotation represent essential properties of an SLC transporter and therefore decided to keep the weights unchanged for these two modalities.

In the revised manuscript, we added more detail on the weighting and integration process to the result section, and a figure panel with the results of the entropy calculations. The addition of this analysis improved the quality and clarity of our integration approach. We thank the reviewer for provoking this analysis.

Changes: added new Figure panel EV5D [page 32], adapted results section [page 10, lines 414-417] and added the entropy analysis to the methods section [page 38, lines 959-961]. Minor adaptation of the discussion section on the integration weights [page 14, lines 591-593].

4. Ensuring the validity of multimodal data: How can the validity of the integrated multimodal data be ensured to prevent noise? Can an ablation study be performed to assess the contribution of each modality and validate the effectiveness of the data integration?

We thank the reviewer for the critical questions and suggestions in points 4, 5 and 6, all regarding the multimodal integration and clustering. As a ground-truth of functional SLC-SLC similarity is missing, we can only rely on the functional annotation of SLC properties (fold, family, substrate classification, subcellular localization) as best proxy of biological function. Using this approximation, we performed two new analyses to quality control our integration approach.

- a) To assess the validity and performance of integrating different modalities, we tested the correspondence of the individual distance measures as well as the integrated distance measure with the annotated SLC properties using ANOSIM (Clarke, 1993). While it was to be expected that the structural distance and substrate-based distance will best represent the fold annotation and substrate classification, respectively, each of the eight distance measures showed significant coherence to at least one of the annotated SLC properties, indicating that all of them contribute functional information. The integrated graph-based distance showed a strong and balanced representation of all three SLC properties.
- b) To validate the effectiveness of the clustering on the multimodal integration, we performed a leave-one-out cross-validation of building and clustering the functional landscape. For each of the resulting landscapes, we assessed the enrichments of functional SLC properties within its clusters, as we did in previous Figure EV5G. The full model, representing all eight modalities, showed the highest number of enrichments overall and per SLC property. Removing any of the modalities resulted in a loss of capturing SLC properties in specific

clusters. Ranking the left-out modalities by the number of enrichments lost could be interpreted as contribution to the overall functional landscape, but only with respect to the SLC function represented in the chosen reference properties (fold, family and substrate classification) and not to other functional aspects that we aim to capture with our integrative approach, such as for example protein-protein interactions, subcellular localization or effect on metabolic state.

Despite missing a ground-truth on SLC-SLC similarity and SLC function, we think that the ANOSIM analysis on the distances and the cross-validation analysis on the clustering show that each of the modalities chosen contributes relevant information and that the integrative clustering approach is effectively capturing actual functional similarity.

Changes: Added the two new analyses as panels F and J to Figure EV5 [page 32], to the results section [page 10, lines 424-428; page 11, lines 472-475] and the method section [page 39, lines 980-994].

5. Validating the effectiveness of the multimodal clustering method: Can an experiment be designed to validate the effectiveness of the multimodal clustering method, rather than relying solely on descriptive analysis?

We think that the cross-validation analysis described in the previous point is nicely validating the effectiveness of the multimodal clustering method. Experimental validation requires deorphanization of an SLC. Therefore, new studies discovering novel functional aspects for an SLC will in addition allow to validate the effectiveness retrospectively.

Changes: added point on validation to the discussion section [page 14, lines 601-602].

6. Noise in multimodal clustering: Given the introduction of multiple modalities, is there a risk of noise being generated in the clustering results due to inconsistencies or redundancies between the data types? How can the accuracy of the results be ensured?

Given the correlation analysis in Figure EV5C, we do not expect much redundancy between data types but rather more inconsistency, which we interpret as orthogonal data accommodating different aspects of SLC function. Noise of the input might still be an issue, but will be somewhat reduced by averaging the similarities, at the cost of some information loss. This is described in the discussion section [page 14, lines 593-601].

The new cross-validation analysis introduced above demonstrates that the landscape clustering reflects actual biological functional and is not driven by noise or other artifacts.

7. Comparison with existing methods: Is it necessary to compare this method with existing clustering approaches to assess its relative advantages and limitations?

We thank the reviewer for this suggestion. We indeed tried different integration approaches, such as MOFA, MultiMAP, SNF and AlignedUMAP. Integration with any of these methods led to significant loss of either modalities or number of SLCs, due to inability of handling missing values or due to strong assumptions on data distributions and incompatibility with qualitative data.

Point-by-point response for the manuscript: Data- and knowledge-derived functional landscape of human solute carriers.

We discuss those methods and their limitations in the introduction [page 3, lines 110-119], but we do not think a general comparison and discussion of integration and clustering approaches for complex, high-dimensional datasets is feasible in the context of this work.

11th Mar 2025

Manuscript Number: MSB-2024-12692R

Title: Data- and knowledge-derived functional landscape of human solute carriers

Dear Dr. Superti-Furga,

Thank you for the submission of your revised manuscript to Molecular Systems Biology. We have now received the enclosed reports from the referees that were asked to re-assess it. As you will see the reviewers are now globally supportive and I am pleased to inform you that we will be able to accept your manuscript pending the following final amendments and appropriate response to reviewers:

- 1) Please submit a final version of the manuscript without track changes.
- 2) Please format the Data availability section according to the example below:
"The datasets and computer code produced in this study are available in the following databases:
- Chip-Seq data: Gene Expression Omnibus GSE46748 (<https://www.ncbi.nlm.nih.gov/geo/query/acc.cgi?acc=GSE46748>)
- Modeling computer scripts: GitHub (<https://github.com/SysBioChalmers/GECKO/releases/tag/v1.0>)
- [data type]: [full name of the resource] [accession number/identifier] ([doi or URL or identifiers.org/DATABASE:ACCESSION])"
- 3) Hopefully by now BiImage Archive has validated your submission of the immunofluorescence images associated with your manuscript. Please provide the accession number and link specifically for the accession in the Data Availability statement.
- 4) Please rename "Conflict of Interest" to "Disclosure and competing interests statement". We updated our journal's competing interests policy in January 2022 and request authors to consider both actual and perceived competing interests. Please review the policy <https://www.embopress.org/competing-interests> and update your competing interests if necessary.
- 5) Our journal encourages inclusion of *data citations in the reference list* to directly cite datasets that were re-used and obtained from public databases. Data citations in the article text are distinct from normal bibliographical citations and should directly link to the database records from which the data can be accessed. In the main text, data citations are formatted as follows: "Data ref: Smith et al, 2001" or "Data ref: NCBI Sequence Read Archive PRJNA342805, 2017". In the Reference list, data citations must be labeled with "[DATASET]". A data reference must provide the database name, accession number/identifiers and a resolvable link to the landing page from which the data can be accessed at the end of the reference. Further instructions are available at .
- 6) Data not shown: We do not allow statements/conclusions with "data not shown". All data referred to in the paper should be displayed in the main or Expanded View figures. Please remove from page 16.
- 7) In the Methods, please take care of the following:
 - The Material and Methods section should be renamed to "Methods".
 - Cell lines: Please be sure to include a sentence in the Methods as to whether or not the cell lines were recently authenticated and tested for mycoplasma contamination. Please also be sure to update the Author Checklist with this information and where it can be found in the manuscript.
 - Please ensure that a statement on whether or not blinding was done is included in the Methods even if no blinding was done. Please also be sure to update the Author Checklist with this information and where it can be found in the manuscript.
- 8) Please place individual sections of the manuscript in the following order: Title page - Abstract & Keywords - Introduction - Results - Discussion - Methods - Data Availability - Acknowledgements - Disclosure and Competing Interests Statement - References - Figure Legends - Expanded View Figure Legends.
- 9) For the figures and figure legends, please take care of the following:
 - Please remove all figures from main manuscript file and leave only main figure legends placed after the references.
 - Please note that the exact p values are not provided in the legend of figure EV5 F
 - Please note that the box plots need to be defined in terms of minima, maxima, centre, bounds of box and whiskers, and percentile in the legend of figure 1C.
- 10) Dataset EV: The legends for all EV datasets should be removed from the main manuscript file and uploaded as an individual tab/sheet in each corresponding Excel file.
- 11) Table EV: Labels and legends for Tables EV1 and EV2 should be included in each corresponding file, and the legends should be removed from main manuscript file.
- 12) Please remove the 'Supplementary Information' section in the main manuscript file.
- 13) As part of the EMBO Publications transparent editorial process initiative (see our policy here: https://www.embopress.org/transparent-process#Review_Process), Molecular Systems Biology will publish online a Peer Review File (PRF) to accompany accepted manuscripts. This file will be published in conjunction with your paper and will include the anonymous referee reports, your point-by-point response and all pertinent correspondence relating to the manuscript. Let us know whether you agree with the publication of the PRF and as here, if you want to remove or not any figures from it prior to publication. Please note that the Authors checklist will be published at the end of the PRF.
- 14) After your paper is published, we will promote it on social media. If you have any handles or hashtags for Bluesky you would like included, please let us know.
- 15) Please provide a point-by-point letter INCLUDING my comments as well as the reviewer's reports and your detailed

responses (as Word file).

I look forward to reading a new revised version of your manuscript as soon as possible.

Yours sincerely,

Poonam Bheda, PhD
Scientific Editor
Molecular Systems Biology

Reviewer #1:

The authors have adequately addressed my comments and I think the manuscript is ready for publication.

Reviewer #2:

I thank the reviewers for responding to my comments in a clear manner.

I'm pleased to see that the manuscript has improved substantially in clarity thanks to the edits that the authors have performed.

I have one further small point. In their responses the authors chose to omit a direct comparison between their curation and the curation of Zhang et al., which at the time was a Biorxiv preprint. However, it seems this study has now been published: <https://pubmed.ncbi.nlm.nih.gov/39930358/>, and it does seem relevant to the manuscript. I would like the authors to include a reference to this and note the agreement between the curation as this is useful for readers to note. If this is done I'd be delighted to see the manuscript published.

Reviewer #3:

Thank you to the authors for their detailed responses to my review comments. After carefully reviewing the revised manuscript, I am pleased to see that the authors have effectively addressed the key concerns I raised. Overall, I recommend accepting the manuscript for publication.

Point-by-point response for the manuscript: Data- and knowledge-derived functional landscape of human solute carriers. MSB-2024-12692RR

Point-by-point response to final amendments for the manuscript "Data- and knowledge-derived functional landscape of human solute carriers"

1) Please submit a final version of the manuscript without track changes.

We removed all the track changes for the final version of the manuscript.

2) Please format the Data availability section according to the example below:

"The datasets and computer code produced in this study are available in the following databases:

- Chip-Seq data: Gene Expression Omnibus GSE46748

(<https://www.ncbi.nlm.nih.gov/geo/query/acc.cgi?acc=GSE46748>)

- Modeling computer scripts: GitHub (<https://github.com/SysBioChalmers/GECKO/releases/tag/v1.0>)

- [data type]: [full name of the resource] [accession number/identifier] ([doi or URL or identifiers.org/DATABASE:ACCESSION])"

We adapted the Data Availability section accordingly.

3) Hopefully by now BioImage Archive has validated your submission of the immunofluorescence images associated with your manuscript. Please provide the accession number and link specifically for the accession in the Data Availability statement.

The imaging dataset has since been validated and assigned an identifier (S-BIAD1630). Unfortunately, it is still not published, but the EBI helpdesk assured us this will happen within a week.

The accession number along with the final link can now be found in the Data Availability section.

4) Please rename "Conflict of Interest" to "Disclosure and competing interests statement". We updated our journal's competing interests policy in January 2022 and request authors to consider both actual and perceived competing interests. Please review the policy <https://www.embopress.org/competing-interests> and update your competing interests if necessary.

This section has been updated accordingly.

5) Our journal encourages inclusion of *data citations in the reference list* to directly cite datasets that were re-used and obtained from public databases. Data citations in the article text are distinct from normal bibliographical citations and should directly link to the database records from which the data can be accessed. In the main text, data citations are formatted as follows: "Data ref: Smith et al, 2001" or "Data ref: NCBI Sequence Read Archive PRJNA342805, 2017". In the Reference list, data citations must be labeled with "[DATASET]". A data reference must provide the database name, accession number/identifiers and a resolvable link to the landing page from which the data can be accessed at the end of the reference. Further instructions are available at <https://www.embopress.org/page/journal/17574684/authorguide#referencesformat>.

After reviewing all our references to external datasets, we confirmed that we consistently refer to entire databases rather than individual records. Therefore, we have decided to maintain the standard referencing format.

6) Data not shown: We do not allow statements/conclusions with "data not shown". All data referred to in the paper should be displayed in the main or Expanded View figures. Please remove from page 16.

Point-by-point response for the manuscript: Data- and knowledge-derived functional landscape of human solute carriers.

We rephrased the sentence to describe what is shown in the figure instead of what is left out and added a reference to the full dataset. The figure legend now reads:

For readability, the plot displays only SLCs that are either exclusively localized or found in combinations of two locations with at least 5 SLCs. The full localization dataset is available in Dataset EV4.

7) In the Methods, please take care of the following:

- The Material and Methods section should be renamed to "Methods".
- Cell lines: Please be sure to include a sentence in the Methods as to whether or not the cell lines were recently authenticated and tested for mycoplasma contamination. Please also be sure to update the Author Checklist with this information and where it can be found in the manuscript.
- Please ensure that a statement on whether or not blinding was done is included in the Methods even if no blinding was done. Please also be sure to update the Author Checklist with this information and where it can be found in the manuscript.

In the Methods section, we renamed the heading, added a statement and references for cell line quality control (mycoplasma testing), as well as a statement on blinding. The Author Checklist has been updated accordingly.

8) Please place individual sections of the manuscript in the following order: Title page - Abstract & Keywords - Introduction - Results - Discussion - Methods - Data Availability - Acknowledgements - Disclosure and Competing Interests Statement - References - Figure Legends - Expanded View Figure Legends.

We adapted the order of sections accordingly and additionally put the Tables section in between the References and Figure Legends sections.

9) For the figures and figure legends, please take care of the following:

- Please remove all figures from main manuscript file and leave only main figure legends placed after the references.
- Please note that the exact p values are not provided in the legend of figure EV5 F
- Please note that the box plots need to be defined in terms of minima, maxima, centre, bounds of box and whiskers, and percentile in the legend of figure 1C.

Figures have been removed from the manuscript and the legends placed after the references section.

We added the exact p-values into Fig. EV5F and adapted its legend.

The legend to Fig. 1C has been extended to include a definition of the box plots used.

10) Dataset EV: The legends for all EV datasets should be removed from the main manuscript file and uploaded as an individual tab/sheet in each corresponding Excel file.

We removed the Supplementary Information section and placed the EV Dataset legends into the corresponding Excel files.

11) Table EV: Labels and legends for Tables EV1 and EV2 should be included in each corresponding file, and the legends should be removed from main manuscript file.

Point-by-point response for the manuscript: Data- and knowledge-derived functional landscape of human solute carriers.

We removed the Supplementary Information section and placed the EV Table legends into the corresponding Word files.

12) Please remove the 'Supplementary Information' section in the main manuscript file.

We removed the Supplementary Information section from the manuscript file.

13) As part of the EMBO Publications transparent editorial process initiative (see our policy here: https://www.embopress.org/transparent-process#Review_Process), Molecular Systems Biology will publish online a Peer Review File (PRF) to accompany accepted manuscripts. This file will be published in conjunction with your paper and will include the anonymous referee reports, your point-by-point response and all pertinent correspondence relating to the manuscript. Let us know whether you agree with the publication of the PRF and as here, if you want to remove or not any figures from it prior to publication. Please note that the Authors checklist will be published at the end of the PRF.

We fully support EMBO's initiative to increase transparency in the publication process and agree to the publication of the Peer Review File (PRF) alongside our manuscript. There is no need to remove any figures.

14) After your paper is published, we will promote it on social media. If you have any handles or hashtags for Bluesky you would like included, please let us know.

Please include the following Bluesky hashtags:

@supertifurgalab.bsky.social
@giuliosupertifurga.bsky.social
@oeaw.bsky.social
@ihieurope.bsky.social

And if you also do X, please include the following hashtags:

@gsf_lab
@giuliosf
@CeMM_News
@RESOLUTE_IMI
@oeaw

15) Please provide a point-by-point letter INCLUDING my comments as well as the reviewer's reports and your detailed responses (as Word file).

We have included this file and addressed all points.

Reviewer #1:

The authors have adequately addressed my comments and I think the manuscript is ready for publication.

We appreciate the reviewer's time and effort in evaluating our manuscript, and we are grateful for the positive feedback.

Reviewer #2:

I thank the reviewers for responding to my comments in a clear manner.

I'm pleased to see that the manuscript has improved substantially in clarity thanks to the edits that the authors have performed.

I have one further small point. In their responses the authors chose to omit a direct comparison between their curation and the curation of Zhang et al., which at the time was a Biorxiv preprint. However, it seems this study has now been published: <https://pubmed.ncbi.nlm.nih.gov/39930358/>, and it does seem relevant to the manuscript. I would like the authors to include a reference to this and note the agreement between the curation as this is useful for readers to note.

If this is done I'd be delighted to see the manuscript published.

We appreciate the reviewer's positive feedback and are delighted that our revisions have substantially enhanced the manuscript's clarity. We conducted the requested comparison of substrate annotations and incorporated the reported annotations for SLCs not covered in our study into *Dataset EV2*. This enhancement in data comprehensiveness has certainly improved its utility as a resource for readers.

In the manuscript, we adapted the corresponding paragraph in the Results section, which now reads as follows:

Compared to our previously published annotation (Meixner et al, 2020), 23 SLCs now have substrates assigned and another 13 SLCs with previously annotated substrates were now classified as orphans since existing evidence did not fulfil our set criteria (Fig. EV1D). A recent study (Zhang et al, 2025) annotates substrates for an additional 19 SLCs, although most of these annotations are inferred from orthologs and do not meet our criteria (Methods). For completeness, these additional annotations for a total of 32 SLCs are included in Dataset EV2.

Reviewer #3:

Thank you to the authors for their detailed responses to my review comments. After carefully reviewing the revised manuscript, I am pleased to see that the authors have effectively addressed the key concerns I raised. Overall, I recommend accepting the manuscript for publication.

We sincerely thank the reviewer for the thorough evaluation and positive recommendation. We are pleased to have effectively addressed the concerns raised.

11th Apr 2025

Manuscript number: MSB-2024-12692RR

Title: Data- and knowledge-derived functional landscape of human solute carriers

Dear Dr. Superti-Furga,

Thank you again for sending us your revised manuscript. We are now satisfied with the modifications made and I am pleased to inform you that your paper has been accepted for publication.

Yours sincerely,

Sincerely,

Poonam Bheda, PhD
Scientific Editor
Molecular Systems Biology
